# A functional subset of CD8$^+$ T cells during chronic exhaustion is defined by SIRPα expression

Lara M. Myers[1], Michal Caspi Tal [2], Laughing Bear Torrez Dulgeroff[2], Aaron B. Carmody[3], Ronald J. Messer[1], Gunsagar Gulati[2], Ying Ying Yiu[2], Matthew M. Staron[3,9], Cesar Lopez Angel[4], Rahul Sinha[2], Maxim Markovic[2], Edward A. Pham[2,5], Benjamin Fram[5], Aijaz Ahmed[5], Aaron M. Newman [2,6], Jeffrey S. Glenn[5,7], Mark M. Davis [7], Susan M. Kaech[8,10], Irving L. Weissman [2] & Kim J. Hasenkrug [1]

Prolonged exposure of CD8$^+$ T cells to antigenic stimulation, as in chronic viral infections, leads to a state of diminished function termed exhaustion. We now demonstrate that even during exhaustion there is a subset of functional CD8$^+$ T cells defined by surface expression of SIRPα, a protein not previously reported on lymphocytes. On SIRPα$^+$ CD8$^+$ T cells, expression of co-inhibitory receptors is counterbalanced by expression of co-stimulatory receptors and it is only SIRPα$^+$ cells that actively proliferate, transcribe IFNγ and show cytolytic activity. Furthermore, target cells that express the ligand for SIRPα, CD47, are more susceptible to CD8$^+$ T cell-killing in vivo. SIRPα$^+$ CD8$^+$ T cells are evident in mice infected with Friend retrovirus, LCMV Clone 13, and in patients with chronic HCV infections. Furthermore, therapeutic blockade of PD-L1 to reinvigorate CD8$^+$ T cells during chronic infection expands the cytotoxic subset of SIRPα$^+$ CD8$^+$ T cells.

[1] Laboratory of Persistent Viral Diseases, Rocky Mountain Laboratories, Hamilton, MT 59840, USA. [2] Institute for Stem Cell Biology and Regenerative Medicine, Stanford University School of Medicine, Stanford, CA 94305, USA. [3] Research Technologies Branch, Rocky Mountain Laboratories, NIAID, NIH, Hamilton, MT 59840, USA. [4] Deparment of Immunology, Stanford University School of Medicine, Stanford, CA 94305, USA. [5] Department of Gastroenterology and Hepatology, Stanford University School of Medicine, Stanford, CA 94305, USA. [6] Department of Biomedical Data Science, Stanford University School of Medicine, Stanford, CA 94305, USA. [7] Deparment of Microbiology and Immunology, Stanford University School of Medicine, Stanford, CA 94305, USA. [8] Department of Immunobiology, Yale School of Medicine, New Haven, CT 06520, USA. [9] Present address: Foundational Immunology, AbbVie Bioresearch Center, Worcester, MA 01605, USA. [10] Present address: NOMIS Center for Immunobiology and Microbial Pathogenesis, Salk Institute, La Jolla, CA 92037, USA. These authors contributed equally: Lara M. Myers, Michal Caspi Tal. Correspondence and requests for materials should be addressed to K.J.H. (email: khasenkrug@nih.gov)

Key effectors in host immune responses to intracellular pathogens are CD8[+] cytolytic T lymphocytes (CTL). CTLs become activated in a pathogen-specific manner, undergo extensive expansion, and function to locate and kill infected cells. While the destructive capacity of CTLs is essential for their activity, it also provides the potential to cause immunopathological damage[1]. Thus the immune system has evolved multilayered mechanisms to control the duration and magnitude of CTL responses. For example, the contraction of the CD8[+] T cell response is hardwired and not dependent on pathogen clearance[2]. Thus, even in circumstances where a virus is not cleared, the CTL population nevertheless contracts. Furthermore, prolonged antigenic stimulation during chronic infections causes a diminished state of T cell function known as exhaustion[3,4]. Such dysfunction not only protects the host from immunopathology but also contributes to the failure to clear infections[5,6].

T cell exhaustion was first discovered in mice chronically infected with lymphocytic choriomeningitis virus (LCMV)[3,7], but it is now known to also occur in humans chronically infected with viruses such as human immunodeficiency virus (HIV) and hepatitis C virus (HCV)[8]. Exhausted CD8[+] T cells have increased expression of co-inhibitory receptors whose breadth and level of expression have been correlated with dysfunction[9]. Thus high expression of multiple co-inhibitory receptors is considered a cardinal feature of exhausted CD8[+] T cells[6]. Blockade of one of these, programmed cell death protein 1 (PD-1), increases the function of exhausted CD8[+] T cells[10,11]. Cells with intermediate rather than high expression levels of PD-1 have been reported to comprise a subset of less exhausted cells whose function can be rescued by PD-1 blockade[12]. Furthermore, simultaneous blockade of more than one co-inhibitory receptor (e.g., PD-1 and LAG-3[9] or PD-1 and TIM-3[13]) has a much more potent effect on enhancing CD8[+] T cell function than blockade of a single receptor. Thus the state of CD8[+] T cell exhaustion is reversible[14] and evidence indicates that not all CD8[+] T cells become exhausted. Despite their reduced function, exhausted T cells are not uniformly inert and help maintain control over virus replication during chronic infection[15].

In this study we examine the expression of a novel cell surface marker, signal-regulatory protein alpha (SIRPα), expressed on exhausted CD8[+] T cells during chronic infection of mice with Friend virus (FV), a naturally occurring retrovirus of mice[16]. Like other chronic viral infections, chronic FV is associated with exhausted CD8[+] T cells because of sustained antigenic stimulation and suppression by regulatory T cells[17,18]. To identify cell surface markers that might be useful for the identification and therapeutic targeting of unique CD8[+] T cell subsets, we analyzed a publicly available microarray database from CD8[+] T cells isolated from mice chronically infected with LCMV Clone 13 (Cl13)[19] looking for transcripts that showed similar expression patterns to the co-inhibitory receptor, PD-1. Interestingly, we found that the expression pattern of SIRPα closely followed that of PD-1.

SIRPα (SHPS-1, CD172a)[20] is an inhibitory receptor whose expression was previously thought to be limited to myeloid cells, hematopoietic stem cells, and neurons[21]. The binding of macrophage SIRPα to its widely expressed ligand, CD47, induces an inhibitory signal for phagocytosis, a "don't eat me" signal[21] that prevents the phagocytosis of healthy cells. Mice with genetic inactivation or mutation of SIRPα have numerous abnormalities, including impairment of phagocyte migration[22], dendritic cell (DCs) homeostasis[23], bone cell differentiation[24], kidney function[25], and interleukin (IL)-17 and interferon (IFN)-γ production[26]. Phagocytes from SIRPα mutant mice also have enhanced respiratory bursts[27]. Cancer cells upregulate CD47 to evade macrophage clearance by inhibiting phagocytosis[28,29]. Positive roles for SIRPα have also been described including a mechanistic role in the fusion machinery of macrophages[30] and the binding of antigen-presenting cells to bovine CD4[+] T cells during priming[31].

Unexpectedly, we found that SIRPα expression was inducible on a subset of CD8[+] T cells during immune activation and that its expression was coincident with PD-1 expression but more limited. Based on its role as a co-inhibitory receptor on macrophages and its expression on PD-1[hi] CD8[+] T cells, we expected that SIRPα might play an inhibitory role in exhausted T cells. Indeed, the SIRPα[+] subset had high expression of inhibitory molecules, but this was counter-balanced by high expression of co-stimulatory molecules. Furthermore, the SIRPα[+] subset had high levels of cytotoxic granules, displayed evidence of recent cytolytic activity (CD107a[+]), and were more cytotoxic ex vivo than the SIRPα[−] subset. In vivo CTL experiments indicated that SIRPα interactions with CD47 were important for optimal cytolytic activity. Thus SIRPα marks the subset of PD-1[+] CD8[+] T cells that retains antiviral activity during chronic FV infection.

## Results

### SIRPα is expressed on CD8[+] T cells during LCMV infection.
To identify cell surface markers that could mark unique subsets of exhausted CD8[+] T cells, an analysis of publicly available microarray data was performed on T cell receptor (TCR) transgenic LCMV-specific CD8[+] T cells that had been adoptively transferred into wild-type (WT) mice infected with either the Armstrong (Arm) strain of LCMV (causes only acute infections) or the Cl13 strain (progresses to chronic infections)[19]. We identified Sirpα as a gene of interest because it showed an expression pattern similar to PD-1 over time and had sustained upregulation during Cl13 chronic infection compared to more transient expression with Arm infection (Fig. 1a, b). Twenty thousand seven hundred and seventy-six genes were analyzed for correlated expression with Pdcd1 and Sirpα ranked in the 97th percentile during acute and chronic infection (Supplementary Fig. 1). Sirpα was of special interest because it had been shown to be important in innate immunity but was not known to be expressed on CD8[+] T cells or other adaptive immune cells. Furthermore, the sustained expression of Sirpα on CD8[+] T cells late after infection with Cl13 suggested that it might identify an interesting subset of cells during exhaustion. Protein expression was verified by flow cytometry on LCMV-specific, transgenic CD8[+] T cells at 42 days post-infection when CD8[+] T cell responses to Arm would have contracted but responses to Cl13 would be largely exhausted and express PD-1. Over 90% of the transgenic CD8[+] T cells remaining after Arm infection were PD-1 low and SIRPα[−] (Fig. 1c). In contrast, over 95% of the transgenic CD8[+] T cells remaining after Cl13 infection were PD-1 high and a significant subset expressed SIRPα (Fig. 1d). The mean fluorescence intensity of SIRPα expression was significantly higher on the CD8[+] T cells responding to the chronic Cl13 strain compared to acute Arm (Fig. 1e).

### SIRPα upregulation during acute and chronic FV infection.
To further examine SIRPα expression on CD8[+] T cells during chronic infection, we analyzed naive controls (Fig. 1f) and mice infected with FV[16] during early acute infection (7 days post-infection (dpi) (Fig. 1g), late acute infection when T cell responses peak (14 dpi) (Fig. 1h), and chronic infection (>6 wpi) (Fig. 1i) when T cells are exhausted[17]. FV-specific CD8[+] T cells were stained with dextramers specific for the immunodominant CD8[+] T cell epitope, gagL[32,33], and with the activation marker CD11a[34] (Fig. 1f–i). Subpopulations gated for these markers as indicated by quadrants with arrows were then analyzed for the expression of PD-1 and SIRPα. Consistent with previous reports, almost all CD8[+] T cells from naive mice were SIRPα[−] and did not stain

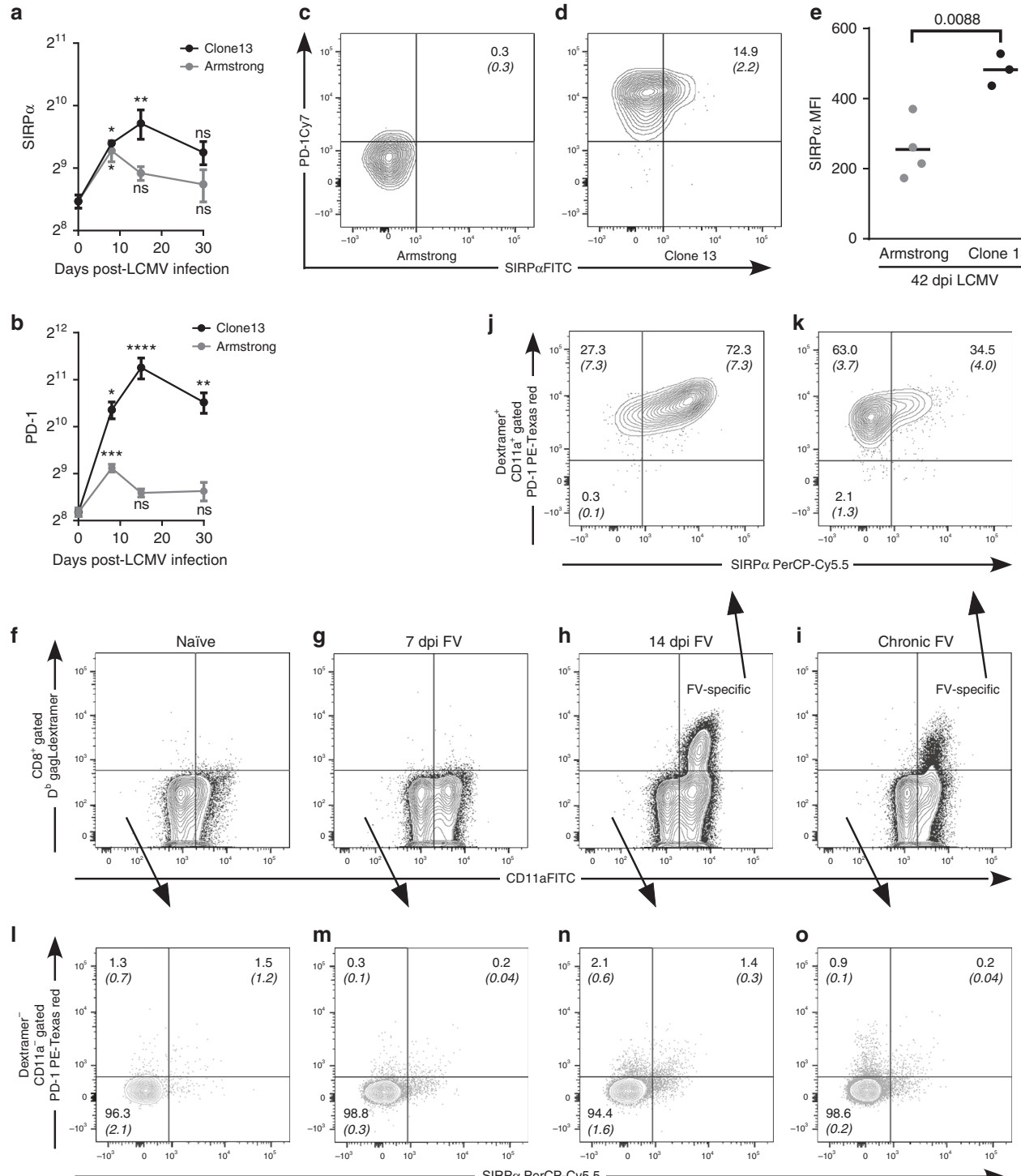

**Fig. 1** Programmed cell death protein 1 (PD-1) and signal-regulatory protein alpha (SIRPα) expression on CD8+ T cells during acute and chronic infection. Wild-type C57/BL6 mice were adoptively transferred with 1000 T cell receptor transgenic lymphocytic choriomeningitis virus (LCMV)-specific CD8+ T cells from the spleens of naive P14 mice and then infected with either LCMV Arm or Cl13. The cells were then analyzed at multiple time points by microarray and the data were made publicly available[19]. SIRPα (**a**) and PD-1 (**b**) expression were analyzed by Dunnett's multiple comparisons test with each time point compared to time zero (n = 4 mice per time point except for d6 LCMV Arm, n = 3. SEMs are shown as bars). Representative flow cytometric contour plots of Thy1.1-gated, adoptively transferred P14 CD8+ T cells at 42 days postinfection with Arm (**c**) or Cl13 (**d**) are shown. Numbers in the upper right quadrant are mean percentages of SIRPα+ cells (n = 4 Arm, n = 3 Cl13), P = 0.0029 by unpaired, two-way t test. Average mean fluorescence intensity of SIRPα expression (P = 0.0088 by unpaired two-way t test) (**e**). CD8+ splenocytes from naive (**f**), 7 dpi (**g**), 14 dpi (**h**), or chronic (**i**) Friend virus (FV)-infected mice were analyzed by flow cytometry for CD11a expression and FV-D[b] gagL dextramer staining. A representative flow cytometry plot is shown. Dextramer+ CD11a+ (**j**, **k**) and dextramer−CD11a− subsets (**l–o**) were further analyzed for PD-1 and SIRPα expression during the course of FV infection. Arrows originate in the quadrant further analyzed and point to the analysis. The percentage in each quadrant depicts the means from eight mice at each time point, with standard deviations in parentheses. The flow cytometric gating strategy is shown in supplementary Fig. 6a–d. Not significant (ns), p > 0.05, *p ≤ 0.05, **p ≤ 0.01, ***p ≤ 0.001, ****p ≤ 0.0001 (unpaired, two-way t tests)

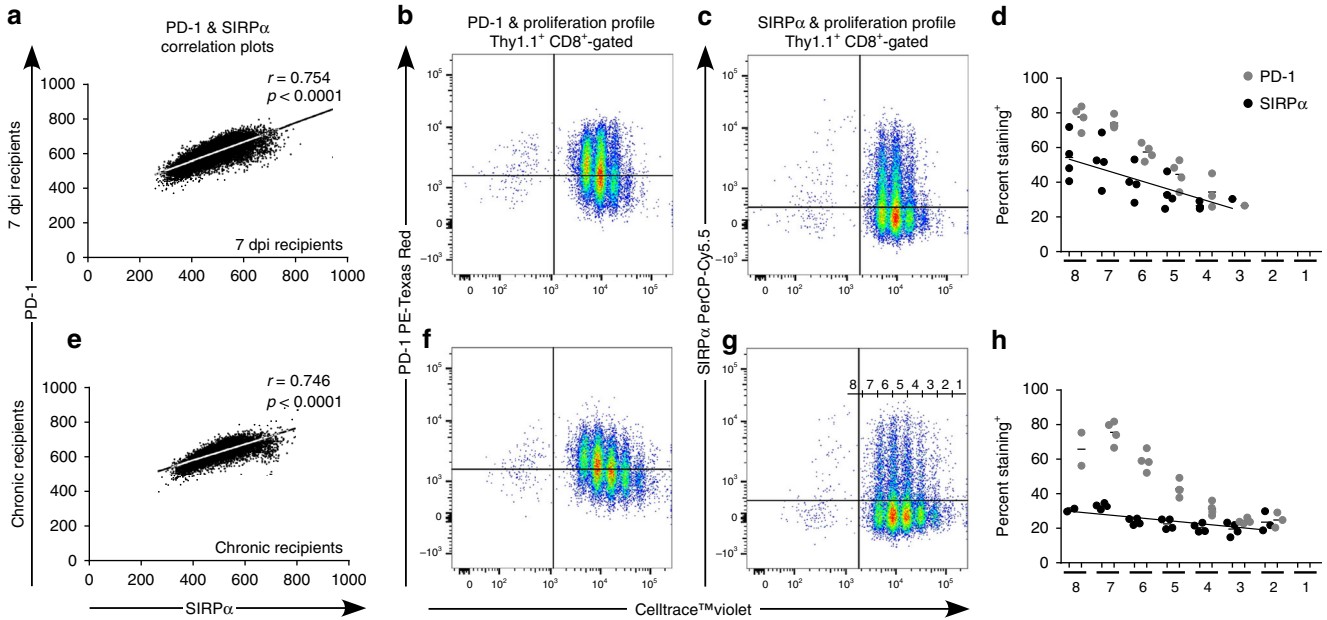

**Fig. 2** Programmed cell death protein 1 (PD-1) and signal-regulatory protein alpha (SIRPα) expression kinetics during Friend virus (FV) infection. Day 7 acute or chronically infected mice were adoptively transferred with 1 × 10⁶ bead-purified and CellTrace™ (violet)–labeled CD8⁺ T cells from the spleens of naive Thy1.1⁺ CD8.TCR transgenic mice. The spleens of recipient mice were analyzed by flow cytometry at 72 h post-transfer. (**a**, **e**) Analysis of dual expression of PD-1 and SIRPα by Pearson correlation showed highly significant correlation ($P < .0001$ for both actue and chronic). Representative plots showing the expression of PD-1 (**b**, **f**) and SIRPα (**c**, **g**) during proliferation (analyzed by dilution of CellTrace™ fluorescence) is shown on donor cells from acute and chronic recipients. Quantification of results from individual mice during acute (**d**) and chronic infection (**h**) are shown with each dot representing an individual mouse and the bar representing the mean. The 1–8 designation depicts the gating strategy for identifying individual cell divisions, but the numbers are only relative as the zero division expression of CellTrace was not evident. Data are from one of two independent experiments with similar results. Linear regression analyses were used to draw the lines in **d** ($R^2 = 0.6148$, $P = 0.0369$) and **h** ($R^2 = 0.8837$, $P = 0.0053$). The difference between the slopes of the lines for acute ($m = 5.781$) vs chronic ($m = 1.836$) was very significant, $P = .0088$. During transfer into chronically infected mice (**h**), there was a slight but significant increase in the proportion of cells expressing SIRPα between cell division 2 (mean = 23.5) and cell division 7 (mean = 32.83), $P = 0.0241$ by two-way Student's $t$ test. The flow cytometric gating strategy is shown in supplementary Fig. 6e–g

with FV-specific dextramers (Fig. 1f, l). At 7 dpi, there were still very few dextramer⁺ cells, but by 14 dpi there was a distinct subpopulation of activated, dextramer⁺ cells (Fig. 1h) that expressed PD-1 and a large majority of which (mean = 72.3%) also expressed SIRPα (Fig. 1j) albeit at lower levels than macrophages (Supplementary Fig. 2). During chronic infections, dextramer⁺ cells were preserved (Fig. 1i), expressed PD-1 (Fig. 1k), and about one third of them also expressed SIRPα (mean = 34.5%) (Fig. 1k). Cells high in SIRPα expression were generally also high in PD-1 expression (Fig. 1j, k). SIRPα was also expressed on activated (CD11a⁺) CD8⁺ T cells responding to other FV peptides (Supplementary Fig. 3). Thus SIRPα was expressed on activated CD8⁺ T cells during both acute (Fig. 1j) and chronic FV infection (Fig. 1k and Supplementary Fig. 3) while non-activated cells remained predominantly negative (Fig. 1l–o).

**SIRPα upregulation after cell division**. To examine the kinetics of SIRPα upregulation during FV infection, adoptive transfer experiments were performed using labeled, FV-specific TCR transgenic CD8⁺ T cells[35] carrying the Thy1.1⁺ genetic marker. Naive donor cells were adoptively transferred into Thy1.2⁺ mice that were either acutely (7 dpi) or chronically (>6 wpi) infected with FV. Such cells adoptively transferred into acutely infected mice are highly functional, whereas they rapidly become dysfunctional upon transfer into chronically infected recipients[17]. Three days after transfer, the donor cells were analyzed for the expression of SIRPα, PD-1, and proliferation (dilution of fluorescent signal). Pearson correlation analyses showed highly

significant correlations between the expression of PD-1 and SIRPα in both acutely and chronically infected mice (Fig. 2a, e). Both PD-1 and SIRPα expression were induced during cell division (Fig. 2b, c, f, g) and the expression data were quantified for multiple mice at each cell division (Fig. 2d, h). In chronically infected mice, SIRPα expression was rapidly induced in about 20% of the transferred cells and slightly but significantly increased to about 35% (Fig. 2h), similar to the endogenous subset (Fig. 1k). By contrast, cells transferred into acutely infected mice showed increasing levels of SIRPα expression throughout all divisions (Fig. 2d). Thus donor cells from the same pool of naive SIRPα⁻ cells had much different levels and kinetics of SIRPα induction dependent on whether they were transferred into acutely infected or chronically infected mice.

**Distinct phenotype of SIRPα⁺ CD8⁺ T cells**. To determine whether the FV-specific PD-1/SIRPα double-positive CD8⁺ T cells from chronically infected mice comprised a subset of cells with a distinct phenotype, the expression of additional markers was examined by flow cytometry. The PD-1⁺ SIRPα⁺ CD8⁺ T cells from chronically infected mice also expressed high levels of the co-inhibitory receptors Tim3 and Lag3, CD95 (Fas), which leads to apoptosis upon ligand binding, and IL-2Rβ chain (CD122), which helps drive a PD-1ʰⁱ phenotype[36] (Fig. 3a–d). This expression pattern would suggest an exhausted phenotype except that these cells also expressed high levels of the activation/co-stimulatory molecules, CD43, CD44, CD40, and CD278 (inducible T cell co-stimulator) (Fig. 3e–h). Neither the SIRPα⁻ or the SIRPα⁺ subsets showed high expression of the terminal

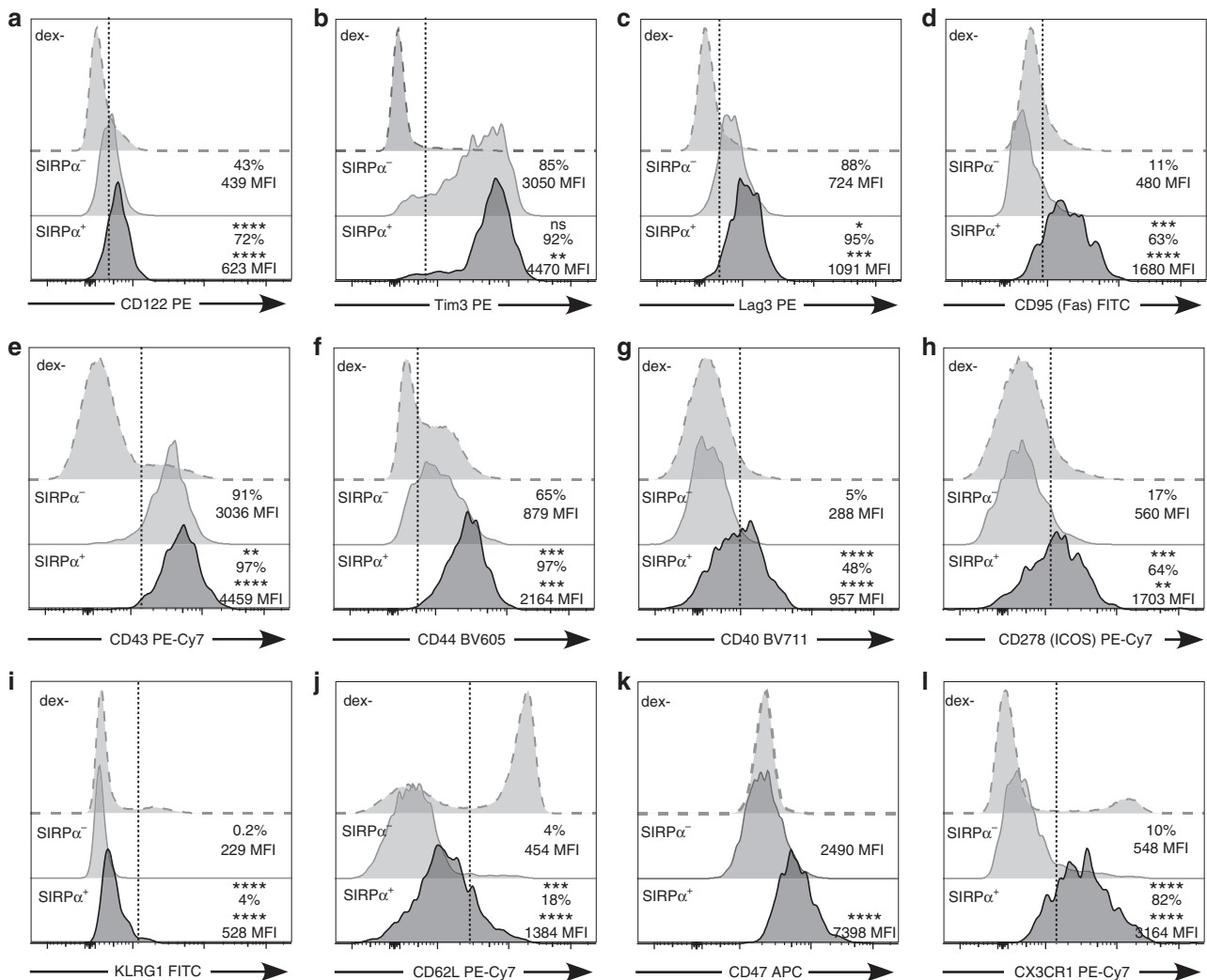

**Fig. 3** Phenotype of Friend virus (FV)-specific PD-1+ SIRPα− and PD-1+ SIRPα+ CD8+ T cells in mice chronically infected with FV. Splenocytes from mice chronically infected with FV were analyzed by multiparameter flow cytometry for surface expression of (**a**) CD122, (**b**) Tim3, (**c**) Lag3, (**d**) CD95 (Fas), (**e**) CD43, (**f**) CD44, (**g**) CD40, (**h**) CD278 (inducible T cell co-stimulator), (**i**) KLRG1, (**j**) CD62L, (**k**) CD47, and (**l**) CX3CR1. A representative off-set histogram overlay is displayed for each marker as well as the average geometric mean fluorescence intensity (MFI) from one experiment is given ($n = 4$ mice). CD8+ dextramer− cells (non-DbgagL-specific cells from infected mice) are shown in dashed gray, PD-1+/SIRPα− CD8+ dextramer+ cells are shown in solid line gray, and PD-1+/SIRPα+ CD8+ dextramer+ are shown in black. The vertical dashed line delineates positivity relative to the FMO control). Results are from one of three independent experiments with similar results (with $n = 8$ additional mice). ns, $p > 0.05$, $*p \le 0.05$, $**p \le 0.01$, $***p \le 0.001$, $****p \le 0.0001$ (unpaired, two-way $t$ tests). The flow cytometric gating strategy is shown in supplementary Fig. 6h–m

differentiation marker, KLRG1, but mean expression was twice as high on the SIRPα+ subset as the SIRPα− subset (Fig. 3i). The expression of CD62L (L-selectin lymphoid homing receptor) on SIRPα+ CD8+ T cells, which is downregulated during activation but returns on memory T cells, was intermediate between SIRPα− CD8+ T cells and naïve CD8+ T cells (Fig. 3j). SIRPα+ CD8+ T cells expressed high levels of CD47, the ligand for SIRPα (Fig. 3k), which is an IFN-inducible gene and its increased expression marks functional, long-lived memory CD4+ T cells[37]. CX3CR1, reported to identify granzyme B positive, cytotoxic memory CD8+ T cells[38,39] was also much higher on SIRPα+ than SIRPα− CD8+ T cells (Fig. 3l). Thus the PD-1+ SIRPα+ CD8+ T cells had a unique surface marker expression phenotype with high expression of both co-inhibitory and co-stimulatory molecules and characteristics of a functional memory phenotype.

**SIRPα+ CD8+ T cells have a unique transcriptional profile**. To gain a broad perspective of the differences between CD8+ T cells

expressing SIRPα or not, whole-transcriptome shotgun sequencing (RNA-SEQ) was performed on cell-sorted populations of splenic SIRPα− and SIRPα+ FV-specific TCR transgenic CD8+ T cells that had been adoptively transferred into FV chronically infected mice 2 weeks earlier. A total of 325 transcripts were differentially expressed at a significant level between the subpopulations, and 82% of the differentially expressed transcripts were upregulated in the SIRPα+ population (Supplementary data 1). Granzyme B (*Gzmb*), Ki-67 (*Mki67*) IFN-γ (*IFNG*), and the inflammatory chemokines *CCL3* and *CCL4* were significantly upregulated in the SIRPα+ subset (Fig. 4, Supplementary data 1), which is consistent with these cells expressing markers of activation and co-stimulation (Fig. 3). Analysis of the top 100 most differentially upregulated genes by gene set enrichment analysis (ToppFun) revealed that the top biological process upregulated by SIRPα+ CD8+ T cells was positive regulation of the immune system followed by active proliferation (Supplementary Table 1). Intriguingly, the next highest biological process was negative immune regulation. Thus the SIRPα+ T cells transcribed

numerous genes capable of both immune activation and inhibitory functions with a skewing toward activation. Comparison of the genes that correlated with SIRPα expression in LCMV-specific CD8[+] T cells with the genes significantly upregulated in FV-specific PD1[+] SIRPα[+] CD8 T cells identified 158 genes that were shared (Supplementary data 2 and Supplementary Figure 4). The most downregulated gene in the SIRPα[+] subset was *Perm1*, an inducer of mitochondrial biogenesis and oxidative phosphorylation[40] typically utilized by exhausted T cells, whereas effector T cells downregulate mitochondrial biogenesis in favor of the glycolytic pathway[41]. Another highly downregulated gene was an inhibitor of T cell activation, Pik3ip1[42]. Thus the phenotyping (Fig. 3) and transcriptional profiling (Fig. 4) results indicated that SIRPα identified a unique subset within the exhausted population of CD8[+] T cells that preserved effector function.

**SIRPα expression associated with in vivo effector function**. To determine whether the function of the SIRPα[+] subset differed from the SIRPα[−] subset, FV-specific (dextramer[+]) CD8[+] T cells from acutely (Fig. 5a) and chronically (Fig. 5b) infected mice were stained for intracellular expression of granzyme B and surface expression of CD107a, an indicator of recent cytolytic activity. Both the SIRPα[+] and SIRPα[−] subsets had cells expressing granzyme B (Fig. 5c–g). Importantly, almost no SIRPα[−] cells expressed CD107a while more than half of the SIRPα[+] cells were CD107a[+], indicating that they had recently undergone exocytosis (Fig. 5c–f, h). Similar results were found from both acutely and chronically infected mice. In acutely infected mice, a large percentage of both the SIRPα[+] and SIRPα[−] subsets had recently proliferated (Ki-67[+]), although the proportion in the SIRPα[+] subset was significantly higher (Fig. 5i). In chronically infected mice, very few SIRPα[−] cells were Ki-67[+], whereas a mean of approximately 35% of the SIRPα[+] subset was Ki-67[+] (Fig. 5i). Thus the SIRPα[+] subset appeared more functional in both cytolytic activity and proliferative capacity than the SIRPα[−] subset, confirming the transcriptional profile results provided by the RNA-SEQ analysis and ToppFun analysis (Fig. 4, Supplementary data 1, and Supplementary Tables 1).

**In vitro CTL killing by SIRPα[+] CD8[+] T cells**. A direct test of cytolytic activity was done using an in vitro killing assay to compare the SIRPα[−] and SIRPα[+] subsets. To obtain sufficient cells for the assay and to avoid stimulating sorted cells by crosslinking with dextramers, we performed adoptive transfers of genetically labeled (Thy1.1[+]), FV-specific, TCR transgenic CD8[+] T cells specific for the immunodominant FV gag peptide[43]. The cells were adoptively transferred into chronically infected recipients where they were allowed to proliferate and become exhausted for 13–15 days. They were then harvested and fluorescence-activated cell sorted into SIRPα[−] and SIRPα[+] subpopulations and co-cultured with either FV-gag peptide-loaded target cells or control cells. As expected for CD8[+] T cells from a chronic infection, the in vitro CTL activity was low but significantly more FV-specific killing was observed with the SIRPα[+] CD8[+] T cells than with the SIRPα[−] subset (Fig. 5j). For comparison, SIRPα[+] and SIRPα[−] CD8[+] T cell effectors taken from acutely infected mice displayed much higher killing frequencies than cells from chronic infections (Fig. 5k), but consistent with the chronic infection results, more killing was observed with the SIRPα[+] subset compared to the SIRPα[−] subset. Thus, during both acute and chronic FV infections, the expression of SIRPα correlated with enhanced cytolytic ability (Fig. 5j, k) and proliferative capacity (Figs. 4 and 5i), suggesting that SIRPα identified cells that sustained an antiviral response during chronic infection. Such a role has been associated with the transcription

factor, T cell factor-1 (TCF-1)[44] and we observed significantly higher intracellular TCF-1 expression in the SIRPα[+] CD8[+] T subset than in the SIRPα[−] subset (Fig. 5l, m).

**CD47[+] targets are more efficiently killed in vivo**. To confirm that cytolytically active CTLs were present in chronically infected mice and to ascertain whether SIRPα was playing a functional role in that activity, an in vivo CTL killing experiment was performed using viral peptide-loaded target cells that either expressed CD47, the ligand for SIRPα, or had a gene inactivation of CD47[45]. Target cells from both WT and CD47 null genotypes, either FV peptide-loaded or control-treated, were differentially labeled with fluorescent stains (Fig. 6a), and all four types of target cells were adoptively transferred at equivalent numbers (Fig. 6b) into naive or chronically infected mice (Fig. 6c, d). Spleens were harvested 6 h after transfer and analyzed by flow cytometry for killing. CD47 null target cells were susceptible to macrophage-mediated phagocytosis regardless of loading with cognate peptide but no virus-specific killing of targets was observed in naive mice as both control targets and peptide-loaded targets remained at the starting ratio of 50:50 (Fig. 6c). In contrast, virus-specific killing was observed in chronically infected mice (Fig. 6d), which was quantified in two separate experiments. In the first experiment, four of the six chronically infected mice tested displayed CD8[+] CTL activity, and the virus-specific killing was significantly greater in WT targets than in CD47 null targets (Fig. 6e). In the second experiment, all 14 mice displayed CTL activity, which was again significantly greater against the WT targets compared to the CD47 null targets (Fig. 6f). Interestingly, compared to uninfected cells from a FV-infected mouse, the infected cells significantly upregulated expression of CD47 (Fig. 7). Thus SIRPα-CD47 ligation was not required for cytolysis in vivo, but it significantly enhanced cytolysis.

**CD8[+] T cells from human HCV patients upregulate SIRPα**. To determine whether SIRPα expression could also be found on human T cells during a chronic viral infection, CD8[+] T cells from healthy controls or patients with chronic HCV (Table 1) were examined using CyTOF, flow cytometry that uses heavy metal ion-tagged antibodies. In CD8[+] T cells from both HCV uninfected and infected patients, the main subset was SIRPα negative (Fig. 8a, b). However, in HCV patients there was a subpopulation of CD8[+] T cells with increased expression of SIRPα (Fig. 8a, b and Supplementary Fig. 5a, b). We analyzed CD57 and CD28 markers because chronic antigenic stimulation of human CD8[+] T cells is associated with the upregulation of CD57 and downregulation of costimulatory CD28. The CD57[+] CD28[−] subset is increased in HCV patients[46]. Although this subset is heterogenous, it is generally associated with a reduced state of function and proliferation[47]. SIRPα expression was significantly higher on CD8[+] T cells from HCV-infected individuals compared to controls in both the functional CD57[−] CD28[+] subset as well as the CD57[+] CD28[−] subset (Fig. 8c). Samples from one patient were also tested by flow cytometry and an example of the data and comparison with SIRPα expression on macrophages is shown (Supplementary Fig. 5c). Thus SIRPα is also expressed on human CD8[+] T cells and is upregulated during chronic HCV infections. SIRPα[+] cells from both CD57[−] and CD57[+] subsets also had higher levels of phosphorylated signal transducer and activator of transcription factor 3 (STAT3), CD244/2B4, and HLA-DR, indicating a higher activation status compared to their SIRPα[−] counterparts (Fig. 8d–f). These results are consistent with SIRPα[+] marking a subset of functional CD8[+] T cells. Furthermore, stimulation of

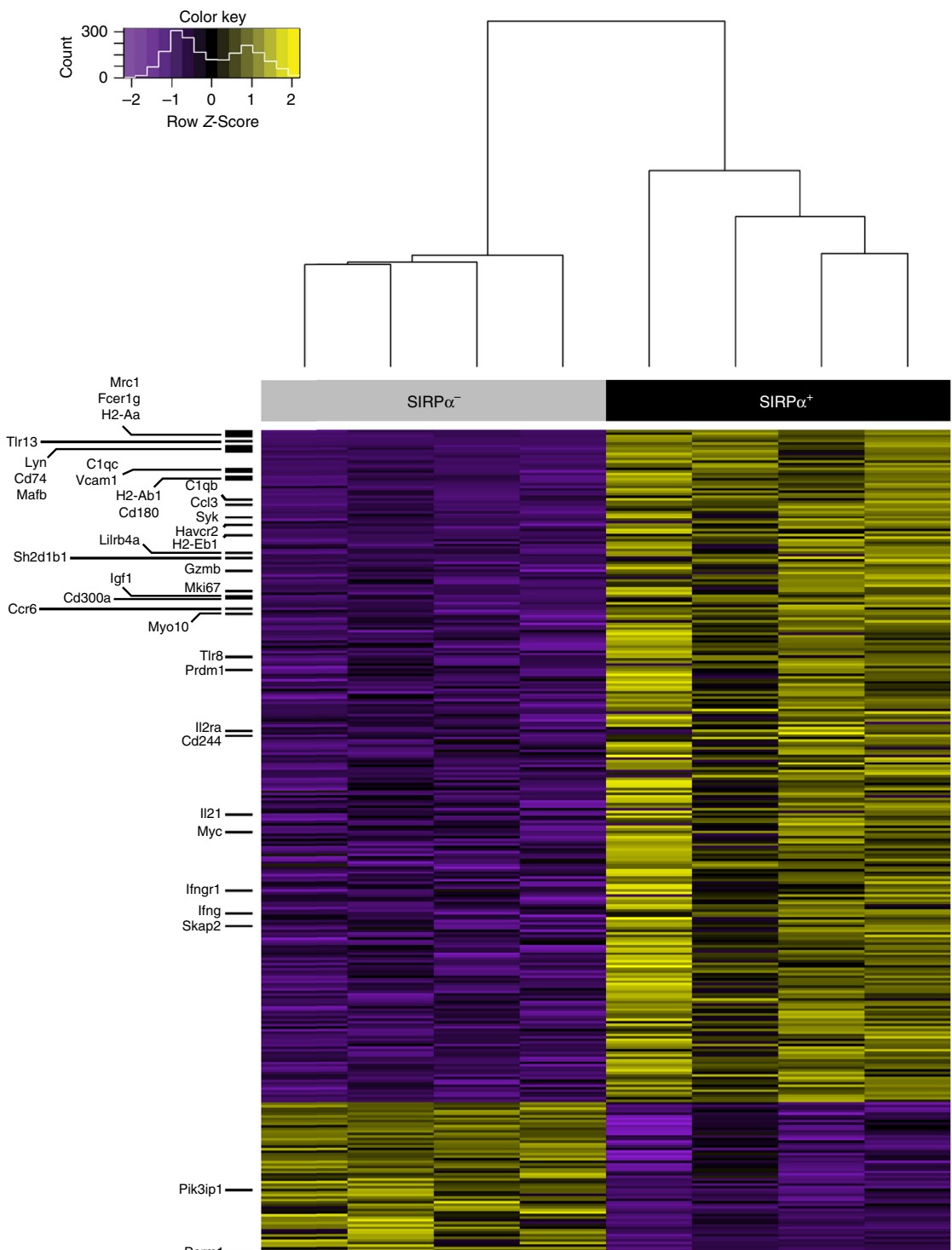

**Fig. 4** Differential gene expression of bulk sorted SIRPα− or SIRPα+ transgenic CD8+PD-1+ T cells. CD8+ T cells from naive Friend virus (FV)-specific Thy1.1+ CD8.TCR transgenic mice were transferred i.v. into Y10 mice chronically infected with FV. After 13–15 days, CD8+ cells were purified from the spleens of these recipients using anti-CD8 paramagnetic beads and the Miltenyi MACS systems. Cells were then stained with anti-Thy1.1; anti-CD8; anti-PD-1; and anti-SIRPα and sorted into CD8+Thy1.1+PD-1+SIRPα− and CD8+Thy1.1+PD-1+SIRPα+ populations for analysis using a BD FACSAriaIlu. A heat map representation of the top 325 differentially expressed genes between the SIRPα− and SIRPα+ transgenic CD8+PD-1+ T cells was generated using the Benjamini–Hochberg procedure to decrease the false discovery rate (FDR) using DESeq2 default settings (FDR ≤ 0.1). Selected differentially expressed genes are highlighted on the left. Heat map is color coded by row z-score as shown

human peripheral blood mononuclear cells (PBMCs) with anti-CD3 and anti-CD28 for 5 days led to significant upregulation of SIRPα on proliferating CD8+ T cells in comparison to unstimulated controls (Supplementary Fig. 5d, e).

**Programmed cell death ligand 1 (PD-L1) blockade expands SIRPα+ CD8+ T cells.** Of interest in treating chronic infections and cancer are immune checkpoint inhibitors, such as anti-PD-1 or anti-PD-L1, which can reinvigorate exhausted T cell

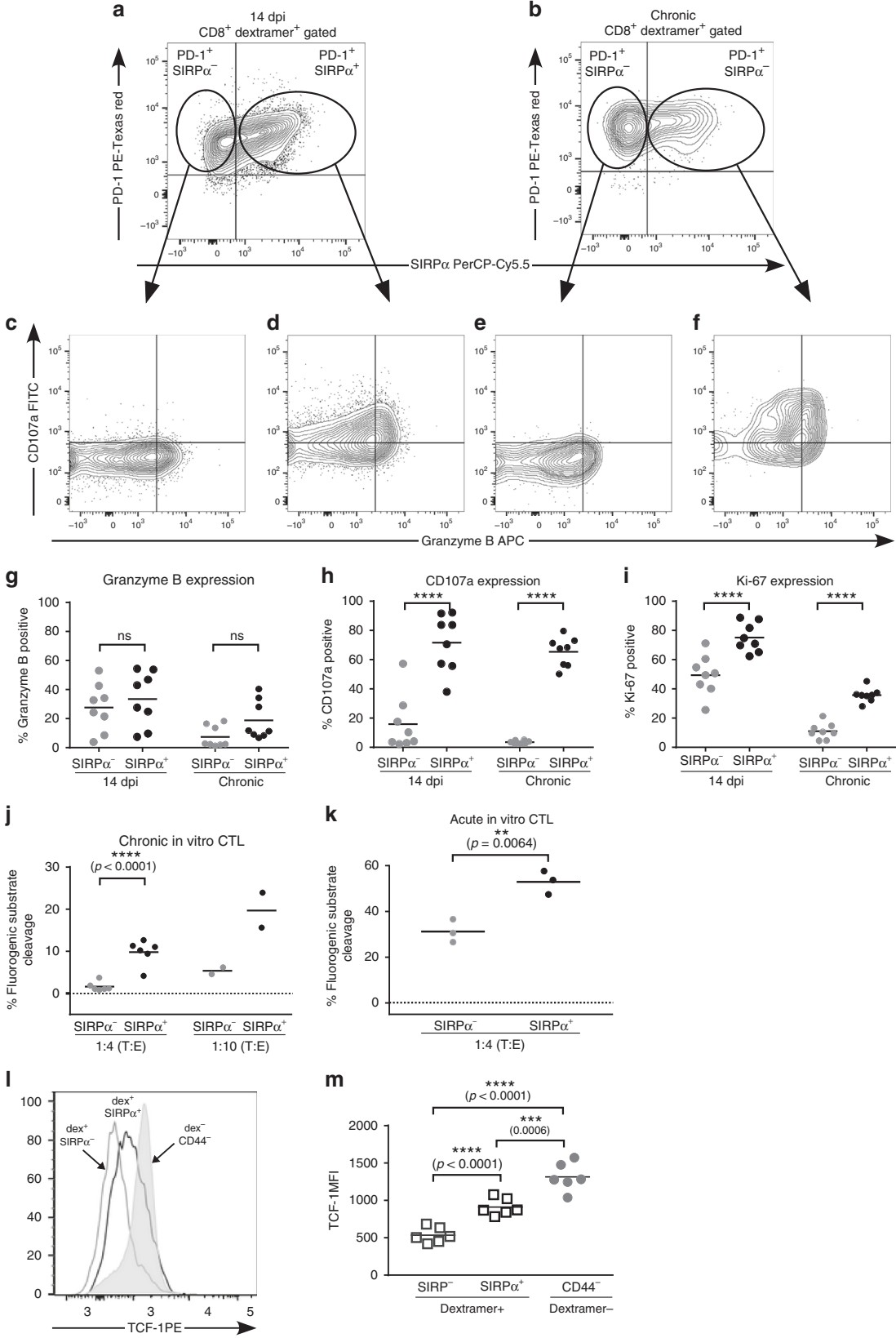

responses. We treated FV chronically infected mice with anti-PD-L1 and observed a significant expansion of FV-specific (Dextramer+) recently cytolytic (CD107a+) CD8+ T cells (Fig. 9a). An average of 80% of the recently cytolytic CD8+

T cells were also SIRPα+ (Fig. 9b), indicating that either the SIRPα+ subset specifically expanded or that the expanded subset of cytolytic cells upregulated SIRPα. Thus the expression of SIRPα can be used to determine whether

**Fig. 5** Signal-regulatory protein alpha (SIRPα) expression on PD-1+ CD8+ T cells identifies cells with enhanced proliferation and cytolytic ability. CD8+ FV-D$^b$ gagL dextramer+ splenocytes from 14 dpi (**a**) or chronically infected mice (**b**) were analyzed by flow cytometry for programmed cell death protein 1 (PD-1) and SIRPα expression and the gated SIRPα-positive and -negative subsets were analyzed for surface CD107a and intracellular granzyme B. Representative fluorescence-activated cell sorting (FACS) plots with gating strategy are shown in **c**–**f**. Data are from two independent experiments ($n = 8$ total mice) where each dot represents the percentage of (**g**) intracellular granzyme B, (**h**) surface CD107a, and (**i**) intracellular Ki-67 from each cell subset. Mice (**j**) chronically or (**k**) acutely infected with Friend virus (FV) were adoptively transferred with $1 \times 10^6$ naive FV-specific T cell receptor transgenic CD8+ T cells and then at 13–15 days post-transfer the cells were recovered and separated into PD-1+ SIRPα− and PD-1+ SIRPα+ subpopulations by FACS cell sorting. A 2-h in vitro cytotoxicity assay was then performed with the sorted effector cells, as described in Methods. Each dot represents the background-corrected value of substrate fluorescence for an individual sample at target to effector ratios, 1:4 and 1:10. The bar is the mean. The negative control is bead purified CD8+ T cells from a naive Y10 mouse and is represented by the horizontal dashed line. Data from chronic infection are from three independent experiments. Subsets of splenic CD8+ T cells from mice chronically infected with FV were analyzed by intracellular flow cytometry for T cell factor-1 (TCF-1). **l** A representative histogram overlay is displayed, depicting the mean fluorescence intensity (MFI) of each labeled subset. **m** Each dot represents an MFI value of intracellular TCF-1 for a given subset. Data are from 1 of 2 independent experiments; a total of $n = 11$ mice were analyzed. All bars in the figure represent the mean. ns = $p > 0.05$, **$p \leq 0.01$, ***$p \leq 0.001$, ****$p \leq 0.0001$ (unpaired, two-way $t$ tests). The flow cytometric gating strategy is shown in supplementary Fig. 6h–m

immune checkpoint inhibitor therapy successfully expanded functional CD8+ T cells.

## Discussion

Until now, SIRPα has been considered to primarily be an inhibitory signaling receptor expressed predominantly on myeloid cells in the hematopoietic compartment[21,48]. The results presented here present a more complex picture, both in terms of cell-specific expression and perhaps the nature of the signaling. We confirm that SIRPα has little or no expression on naive T cells, as previously shown in mice[49], rats[50], and humans[51]. However, we now show that SIRPα is expressed on activated CTL during viral infection. Such expression may have previously been missed owing to examination of only naive cells. In addition to the expanded cell-specific expression profile, it is possible that SIRPα signaling in CD8+ T cells may not be negative. During both acute and chronic FV infections, almost all CD8+ T cells that showed evidence of recent cytolytic activity (CD107a+) were also SIRPα+ (Fig. 5h). Compared to the SIRPα− subset, the SIRPα+ subset was also significantly more proliferative (Ki-67+) (Fig. 5i), expressed higher levels of TCF-1 (Fig. 5m), had higher expression of IFN-γ message (Fig. 4, Supplementary data 1), and transcribed significantly more genes indicative of immune activation (Fig. 4, Supplementary data 1). Furthermore, cell-sorted SIRPα+ CD8+ T cells from chronically infected mice had greater in vitro cytotoxicity than the SIRPα− cells from the same mice (Fig. 5j). However, SIRPα might simply mark the active CTL subset rather than positively regulating CD8+ T cell functions. CTL targets lacking CD47 were nevertheless killed by CTL in vivo, albeit to a significantly lower level than targets expressing CD47 (Fig. 6e, f). These results indicate that SIRPα–CD47 interactions are involved in the cytolytic process but do not address whether SIRPα acts as a positive or negative regulator of functional CTL development. Given the high level of co-stimulatory molecules expressed by SIRPα+ CD8+ T cells (Fig. 3), it is possible that negative signaling by SIRPα could be overcome by positive regulatory signaling. Further studies will be required to determine how SIRPα expression, specifically in CD8+ T cells, impacts development and function. What is most evident and novel from the data is that cell-specific SIRPα expression is not as limited as previously thought, that it is expressed on the most proliferative and functionally active subset of CD8+ T cells in both acute and chronic infections, and that it is involved in the cytolytic process. As such, SIRPα will allow scientists and clinicians to follow the expansion or contraction of the functional subset during immunotherapy with relevance not only to infections but also cancer and autoimmune diseases. We find that SIRPα protein surface expression is increased not only in activated mouse T cells but also in activated human T cells, suggesting that this may be a conserved marker of active CD8+ T cells. The elevated SIRPα levels on CD8+ T cells from patients with chronic HCV infection was most pronounced on the CD57− CD28+ subset.

It is worth considering how SIRPα–CD47 interactions might be involved in the cytolytic process in vivo since cognate target cells that did not express CD47 were killed less effectively than WT target cells in multiple in vivo CTL assays (Fig. 6e, f). One possibility is that SIRPα delivered activating rather than inhibitory signals, which is not unprecedented. It has previously been shown that SIRPα signaling in macrophages could activate the production of nitrous oxide and reactive oxygen species via Janus-activated kinase/STAT and phosphoinositide-3 kinase/Rac1/NOX/H$_2$O$_2$ pathways[52]. SIRPα-mediated positive signaling in macrophages is also required for migration and chemotaxis[22,53]. In addition, T cells from SIRPα mutant mice exposed to autoimmune antigens[54,55] or flagellin[26] showed reduced cytokine production, which was attributed to decreased numbers of DCs in the lymphoid tissues of such mice[23]. However, it may also have been an intrinsic T cell property since expression of SIRPα on T cells was neither known at the time nor tested. Thus SIRPα is capable of delivering activating as well as inhibitory signals depending on the context including the presence of adapter proteins such as Skap2[53] and GRB2[56] and phosphatases such as SHP1[57] and SHP2[58]. No evidence of differential expression of *Shp1* or *Shp2* was found in the RNA-SEQ analysis (data not shown), but *Skap2* transcription was increased 2.8-fold in the SIRPα+ subset (Supplementary data 1). Interestingly, one of the most highly overexpressed genes in the SIRPα+ subset (20× increase, Supplementary data 1) was *Lyn*. Lyn is a tyrosine protein kinase with a role in regulating cell activation, and like SIRPα, it has an inhibitory role in myeloid cells[59]. In B cells, Lyn phosphorylation initiates an activation cascade[60], indicating that like SIRPα it can deliver either inhibitory or activating signals in a context-dependent manner. Alternatively, it is possible that SIRPα–CD47 interactions simply stabilize cell-to-cell contacts and the cytolytic synapse. The spanning distance of end-to-end bound CD47–SIRPα complexes (~14 nm, similar to TCR–MHC, CD28–CD86, and CD40–CD40L) suggests that significant binding between a target and effector cell would take place predominantly in immunological synapses where abundant bulkier cell surface proteins such as CD43 and CD45 that can sterically hinder more short range interactions are redistributed outside of the cytolytic synapse[61]. The strength of the interactions between cells is influenced not only by the affinity between the receptors and ligands but also by their density. Thus the high expression of SIRPα on CTL combined with upregulated CD47 on infected

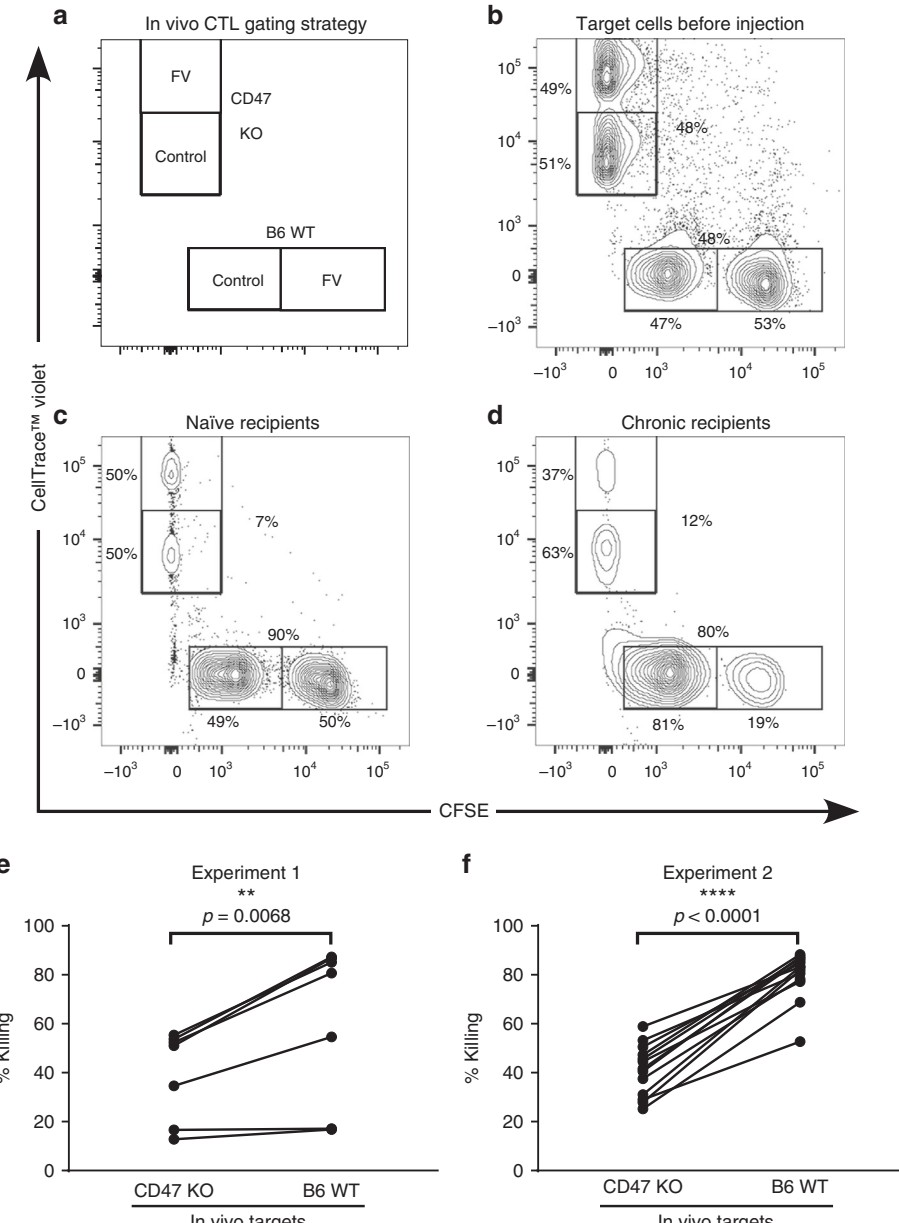

**Fig. 6** Enhanced in vivo cytolytic activity against target cells expressing CD47, the ligand for signal-regulatory protein alpha. Naive and mice chronically infected with Friend virus (FV) were adoptively transferred with differentially fluorophore-labeled target cells as outlined in **a** and in Methods. Briefly, wild-type C57/BL6 splenocytes were differentially labeled with two concentrations of carboxyfluorescein succinimidyl ester and CD47$^{-/-}$ splenocytes were differentially labeled with two concentrations of CellTrace$^{TM}$-violet. The brighter of each subset was peptide-loaded with 25 μM FV-D$^b$GagL peptide, while the lower intensity subset was sham-loaded in dimethyl sulfoxide media. **a** The gating strategy identifying the target cell populations and representative dot plots showing target cell populations **(b)** before injection and retrieved from (**c**) naive and (**d**) chronic recipients 6 h post-transfer. The percentages given are the means for each cell subset combining data from two independent experiments. **e**, **f** The percentage of killing comparing C57/BL6 WT and CD47$^{-/-}$ target cells as described in Methods. The data points showing the killing of each type of target cell within the same recipient mouse were connected with a line and the differences were statistically significant as indicated by two-way paired t tests. Data from the two independent experiments include a total of 20 mice. Virus-specific killing is defined as the percentage of killing of each population of FV peptide-pulsed cells calculated as follows: 100 − ([% peptide pulsed in infected/% un-pulsed in infected)/(% peptide pulsed in uninfected/% unpulsed in uninfected)] × 100). The flow cytometric gating strategy is shown in supplementary Fig. 6n–p

targets (Fig. 7) could have a significant impact on the strength and duration of interactions within the immunological synapse as has been previously suggested[62].

Prior to these experiments, it was known that virus-specific CD8$^+$ T cells were sustained in FV chronically infected mice, albeit at low numbers (~1–3% of CD8$^+$ T cells). It is now apparent that the FV-specific CD8$^+$ T cells in mice with chronic FV are heterogeneous with respect to function and that there is a

cytolytically active subset that can be identified by cell surface expression of SIRPα. As discussed, this active subset displays high expression of both co-stimulatory and co-inhibitory molecules (Fig. 3a–h). Immune checkpoint inhibition by anti-PD-L1 produced an expanded population of CTL, the large majority of which expressed SIRPα (Fig. 9). In chronic LCMV infections the degree of expression of multiple co-inhibitory receptors has been shown to correlate with the severity of T cell dysfunction[6], but

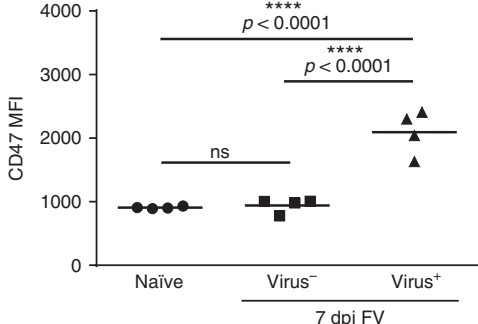

**Fig. 7** Upregulation of CD47 on Friend virus (FV)-infected splenocytes. Mice were infected with FV or left naive and the splenocytes were analyzed by flow cytometry to compare the mean fluorescence intensity of CD47 on naive or FV-infected (Virus+) and uninfected (Virus−) total splenocytes at 7 days post-infection (dpi) with significance as indicated by one-way analysis of variance. Virus was detected by cell surface expression of glycosylated gag antigen with mAb 34 as described in Methods. Data from one of four independent experiments are shown. ns, ****p ≤ 0.0001 (unpaired, two-way t tests)

**Table 1 HCV patient characteristics**

| ID | Previous IFN | Genotype | Liver transplant waitlist | Sex |
|----|--------------|----------|---------------------------|-----|
| 2 | Yes | 1 | No | Male |
| 4 | Yes | 1 | No | Female |
| 7 | No | 2 | No | Female |
| 8 | Yes | 2 | No | Female |
| 9 | Yes | 1 | No | Male |
| 12 | No | 2 | No | Female |
| 13 | No | 1 | Yes | Female |
| 14 | Yes | 1 | No | Male |
| 20 | Yes | 1 | No | Male |
| 22 | Yes | 1 | No | Female |
| 27 | Yes | 1 | No | Male |
| 29 | Yes | 1 | Yes | Male |
| 30 | No | 4 | No | Female |
| 35 | No | 1 | No | Male |
| 38 | Yes | 1 | No | Male |

*HCV* hepatitis C virus, *IFN* interferon

expression of co-inhibitory receptors is not specific to exhausted cells and occurs during T cell activation as well. Thus it is not possible to differentiate dysfunctional T cells from activated T cells based only on the expression of co-inhibitory receptors. Recently, there has been shown to be a great deal of heterogeneity in the level of dysfunction of CD8+ T cells in an exhausted setting such as within a tumor or in a chronic infection[63,64]. A detectable level of function and virus control persists in chronic viral settings as evidenced by the fact that CD8+ T cell-escape variants arise in chronic HIV infections[65,66] and that viral titers increase following depletion of CD8+ T cells in simian immunodeficiency virus-infected macaques[67,68]. Depletion of CD8+ T cells in mice with chronic FV infections does not produce virus relapse, but this is likely due to compensatory mechanisms by antiviral CD4+ T cells[69] and does not indicate that the residual CD8+ T cells exert no control over chronic infection. The SIRPα-positive and -negative CD8+ T cell subsets during exhaustion did not have significantly different expression levels of *Tbet*, *EOMES*, *CTLA4*, or *Bcl2* (Supplementary data 1). Although the SIRPα+ subset displayed higher levels of CD44, Ki67, TCF-1, and CD62L suggesting a functional memory phenotype, they also displayed high

levels of co-inhibitory receptors indicating a different phenotype than previously described for exhausted or functional cells[6].

We have shown that the expression of SIRPα on CD8+ T cells is diagnostic for the presence of active CTLs, even in exhausted settings. Thus SIRPα+ CD8+ T cells are interesting targets for immunotherapy, especially if they can be specifically expanded or further activated to kill chronically infected cells and/or tumors. The fact that target cells without CD47 expression were still targets for CTL killing, albeit to a lesser extent than targets with CD47 expression, is important when considering using CD47 blockade to treat cancer. CD47 is overexpressed by human cancer cells, allowing them to evade macrophage-mediated phagocytosis[29,70,71]. CD47 blockade has been shown to potentiate macrophage-mediated phagocytosis of tumor cells in numerous models[28,72] and is currently in several clinical trials to treat various cancers. The current results suggest that, while activating macrophage antitumor phagocytosis, CD47 blockade might also inhibit CD8+ T cell antitumor activity. However, such inhibition might be overcome using PD-1 blockade, a cancer immunotherapy currently thought to function primarily via CD8+ T cell activation. Interestingly, it was recently shown that, similar to tumor-infiltrating T cells, tumor-associated macrophages also express PD-1 co-inhibitory receptors[73]. Thus cancer therapies targeting PD-1 or its ligand could be activating macrophage functions as well as T cells. A combination of PD-1 and CD47 blockade could have synergistic effects by potentiating the antitumor activity by both macrophages and T cells.

## Methods

**Mice, viruses, infection, and tissue harvest.** For LCMV studies, female 4–6-week-old C57BL/6 J mice from NCI and Thy-1.1+ P14 TCR transgenic mice[74] that recognize the H-2Db gp33 epitope were used where indicated. Mice were intra-peritoneally (i.p.) infected with 2 × 10^5 plaque-forming units (p.f.u.) LCMV Armstrong (Arm)—which causes an acute infection—or intravenously (i.v.) with 2 × 10^6 p.f.u. LCMV Clone 13 (Cl13)—which causes a chronic infection. The use of all animals was conducted in accordance with Yale University IACUC guidelines. For FV studies, mice were female (C57BL/10 × A.BY) F1 (Y10) (H-2^b/b, Fv1^b, Rfv3^r/s, Fv2^r/s) and FV-specific Thy1.1+ CD8.TCR transgenic mice between 12 and 24 weeks of age at the beginning of the experiments and were bred at the Rocky Mountain Laboratories. The FV stock has been passaged in mice for more than three decades and contains three separate viruses: (1) B-tropic Friend murine leukemia helper virus (F-MuLV), which is a replication competent retrovirus; (2) polycythemia-inducing spleen focus-forming virus, which is a defective retrovirus that is packaged by F-MuLV-encoded virus particles; and (3) lactate dehydrogenase-elevating virus, an endemic murine nidovirus related to coronaviruses[75]. Mice were infected by i.v. injection of 0.2 mL phosphate-buffered saline (PBS) containing 1500 spleen focus-forming units of FV complex. Mice were considered chronically infected at 6 weeks postinfection when F-MuLV levels stabilize at approximately 10^4 infectious centers per spleen. Splenocytes were isolated by tissue homogenization through a 100-μm filter and red blood cells were removed using lysis buffer (0.15 M NH_4Cl, 10 mM KHCO_3, 0.1 M EDTA). Mice were treated in accordance with RML IACUC-approved animal use protocols following the regulations and guidelines of the Animal Care and Use Committee of the Rocky Mountain Laboratories and the National Institute of Health Office of Laboratory Animal Welfare.

**LCMV Affymetrix.** Affymetrix arrays from GSE41867 were obtained as CEL files, MAS5 normalized using the "affy" package in Bioconductor, mapped to NCBI Entrez gene identifiers using a custom chip definition file, and converted to MGI gene symbols. Gene expression values were mean-and-log2-normalized prior to analysis.

**Flow cytometry.** For flow cytometric analysis, live lymphocytes were gated using a SSC-A and FSC-A gate. Cells were then gated by time to exclude artifacts caused by erratic sample flow and by FSC-H and FSC-A to exclude doublets. Gating strategies are shown in Supplementary Figure 6. The antibodies used for surface staining were: A700-anti-CD8 (53–6.7, eBioscience 56-0081-82, lot E08952-1633; 1/800) or PacBlue-anti-CD8 (53-6.7, BD Pharmigen 558106, lot 38114; 1/400); fluorescein isothiocyanate (FITC)-anti-CD11a (2D7, BioLegend 101006, lot B165666; 1/400); PE-CF594-anti-PD-1(J43, BD Horizon 562523, lot 7243896; 1/200); PE-Cy7-anti-Thy1.1 (H1551, eBioscience 25-0900-82, lot 4300740; 1/1000); FITC-anti-CD107a (1D4B, BD Pharmigen 553793, lot 02482; 1/100); phycoerythrin (PE)-anti-Tim3 (8B.2C12, eBioscience 12-8571-81, lot E008713; 1/400); PE-anti-Lag3 (C9B7W, BD

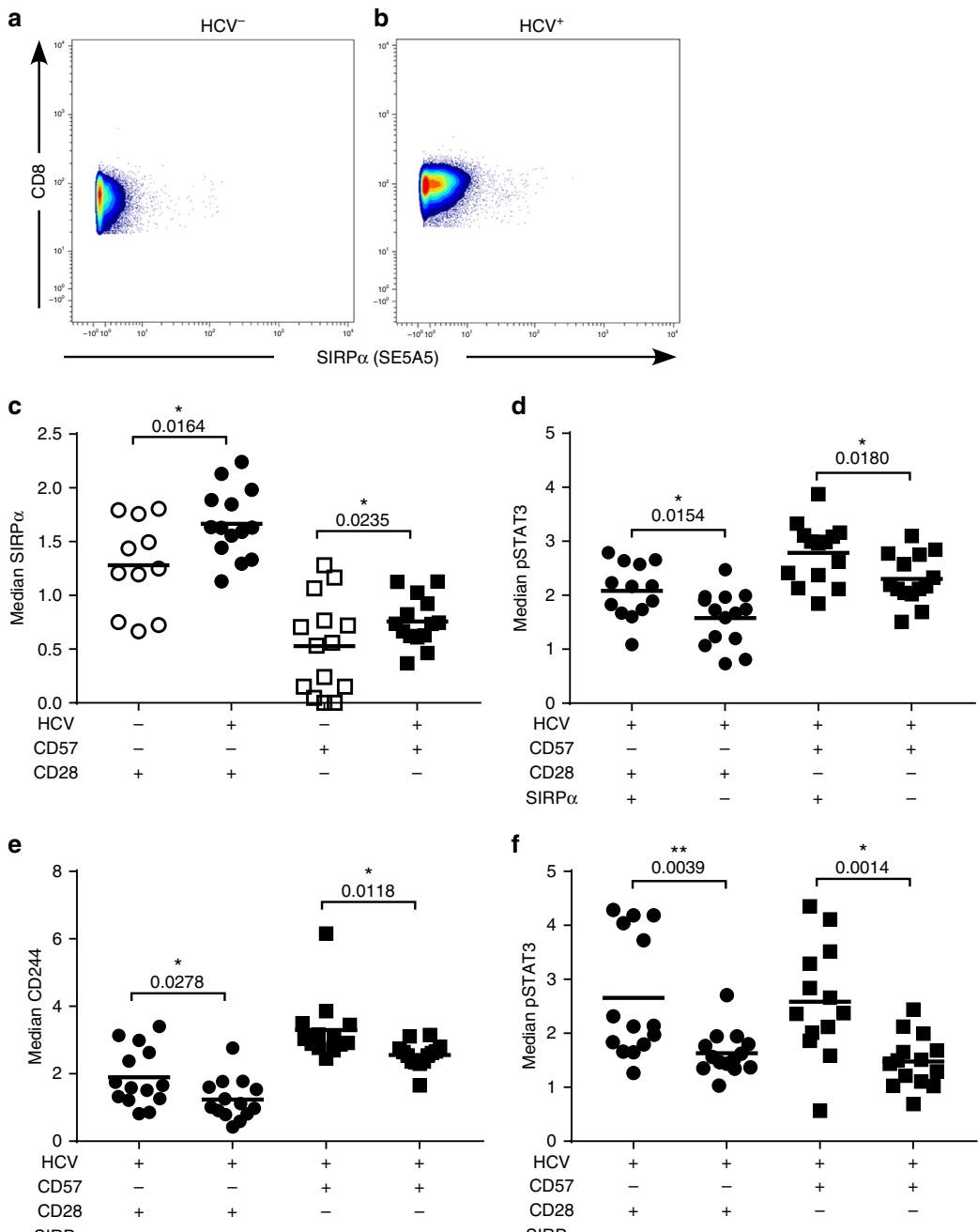

**Fig. 8** Increased signal-regulatory protein alpha (SIRPα) expression on CD8[+] T cells from hepatitis C virus (HCV)-infected patients identifies a more activated phenotype. **a** Representative plots of CyTOF analysis of the median healthy donor and (**b**) the median HCV patient donor in regards to SIRPα expression levels. **c** CD8[+] CD57[−] CD28[+] (circles) and CD8[+] CD57[+] CD28[−] (squares) peripheral blood mononuclear cells from healthy controls (open symbols) and HCV-infected patients (closed symbols) were analyzed by CyTOF for SIRPα expression. For CyTOF analyses, fluorescence intensity data are commonly transformed to arcsinh for analysis and display[81]. The SIRPα[+] and SIRPα[−] subsets of CD8[+] CD57[−] CD28[+] (circles) and CD8[+] CD57[+] CD28[−] (squares) from HCV-infected patients were further analyzed by CyTOF for the expression of (**d**) phosphorylated STAT3, (**e**) CD244, and (**f**) HLADR. Median expression levels (Arcsinh transformed) for each subset are represented by corresponding symbols, where each symbol represents an individual sample and the bar represents the mean. Differences between samples were statistically significant as shown by two-way unpaired t test

Pharmigen 552380, lot 0000054474; 1/50); FITC-anti-Fas (Jo2, BD Pharmigen 15404, lot M045159; 1/100); PE-Cy7-anti-CD43 (1B11, BioLegend 121218, lot B132711; 1/1000); BV605-anti-CD44 (IM7, BD Horizon 563058, lot 7177869; 1/1000); BV711-anti-CD40 (3/23, BD Biosciences 740700, lot 6326576; 1/400); PE-Cy7-anti-CD278 (C398.4A, BioLegend 313520, lot B135805; 1/200); PE-Cy7-anti-CD62L (MEL-14, eBioscience 25-0621-82, lot E07577-943; 1/1000); PE-anti-CD122 (TM-b1, BD Biosciences 553362, lot 24161; 1/200); FITC-anti-KLRG1 (2F1, eBioscience 11-5893-82, lot E09834-484; 1/800); PE-Cy7-anti-CX3CR1 (SA011F11, BioLegend 149016, lot B216575; 1/200); APC-anti-CD47 (miap301, eBioscience 17-0471-82, lot 4301458; 1/100); and PerCP-Cy5.5-anti-SIRPα (P84, BioLegend

144010, lot B252132; 1/100). Monoclonal antibody (mAb) P84 specificity is based on the following: signal regulatory proteins (SIRPα and SIRPβ in the mouse) are expressed on neurons, hematopoietic stem cells, and myeloid cells including macrophages, monocytes, granulocytes, and DCs[76]. DCs, macrophages, and mononcytes from mice with targeted SIRPα gene disruptions completely lose reactivity with mAb p84 (anti- SIRPα) even though their SIRPβ expression is normal. These results indicate specificity of p84 for SIRPα without cross-reactivity for SIRPβ[77]. For FV-specific H-2D[b]/Abu-Abu-L-Abu-LTVFL staining, allophyco-cyanin (APC)- or PE-D[b] gagL-MHC Dextramer (Immudex, Copenhagen, Denmark) was used at 1/25. For intracellular staining, cells were surfaced stained and

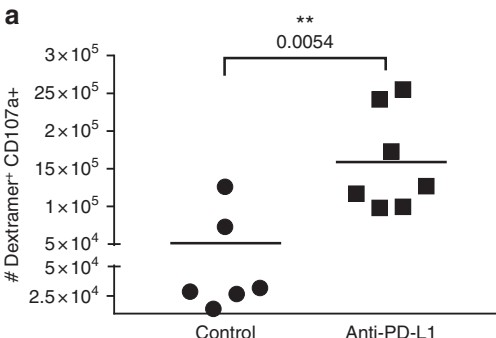
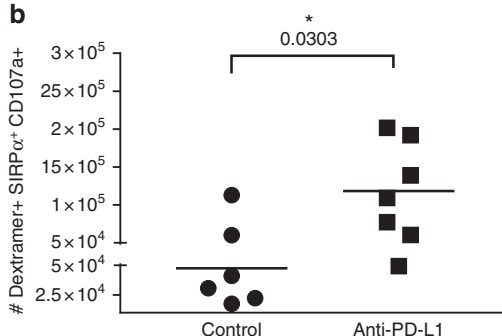

**Fig. 9** Expansion of cytolytic Friend virus (FV)-specific SIRPα+ CD8+ T cells during programmed cell death ligand 1 (PD-L1) blockade. As described in Methods, mice chronically infected with FV were injected every other day with anti-PD-L1 blocking antibody and analyzed the second day following the final injection for the number of dextramer+ CD107a+ (**a**) and dextramer+ SIRPα+ CD107a+ (**b**) T cells in the spleen. Bars represent the mean and symbols represents individual mice (n = 6–7) pooled from 2 independent experiments, with statistical differences analyzed by two-way unpaired t tests

fixed prior to permeabilization. The antibodies used for intracellular staining were PE-anti-EOMES (Dan11mag, eBioscience 12-4875-82, lot E10466-1634;1/200); PE-Cy7-anti-Tbet (eBio4B10, eBioscience 25-5825-82, lot 4277988; 1/200); A700-anti-Ki67 (B56, BD Pharmigen 561277, lot 7073537; 1/200); PE-anti-TCF-1 (S33-966, BD Biosciences 564217, lot 8081983; 1/200), and APC-anti-human granzyme B (GRB05; Molecular Probes Invitrogen GRB05, lot 1908524; 1/50). To stain for intracellular granzyme B, cells were fixed overnight in 0.5% paraformaldehyde (PFA) and then permeabilized with 0.1% saponin/PBS containing 0.1% sodium azide, 0.5% bovine serum albumin, and 50 mM glucose. To stain for all other intracellular markers, intracellular staining was performed using the eBioscience Foxp3 Kit, following the company's recommendation. To detect FV-infected cells, cells were stained with tissue culture supernatant containing mAb (MAb 34), which is specific for F-MuLV glycosylated Gag protein (mAb34 was produced at RML/NIAID/NIH as a culture supernatant). MAb 34 binding was detected with FITC-labeled goat anti-mouse IgG2b (R12-3, BD Pharmigen 553395, lot 32885; 1/800). The muliparameter data were collected with an LSRII (BD Biosciences) and analyzed using the FlowJo software (version 10.2; TreeStar, Inc.).

**T cell adoptive transfers for proliferation and the in vitro CTL assay.** Alpha beta CD8+ T cells from the spleens of naive FV-specific Thy1.1+ CD8.TCR transgenic mice[35] were first isolated by magnetic bead separation (Miltenyi MACS system) following the manufacturer's recommendations. For proliferation assays, the cells were then CellTrace^TM violet-labeled as directed (Invitrogen). A total of 1 × 10^6 CD8+ T cells were transferred i.v. into either acutely infected (7 dpi) or chronically FV-infected recipients. At 72 h post-transfer, the splenocytes were surface stained and then analyzed for CellTrace^TM dilution. For the in vitro CTL assay, CD8+ T cells from naive FV-specific Thy1.1+ CD8.TCR transgenic mice were transferred i.v. into Y10 mice chronically or acutely infected with FV. After 13–15 days, CD8+ cells were purified from the spleens of these recipients using anti-CD8 paramagnetic beads and the Miltenyi MACS system following the manufacturer's recommendations. Cells were then stained with PE-Cy7-anti-Thy1.1; A700-anti-CD8; PE-CF594-anti-PD-1; and PerCPCy5.5-anti-SIRPα and sorted into CD8+Thy1.1+PD-1+SIRPα− and CD8+Thy1.1+PD-1+SIRPα+ populations using a BD FACSAriaIIu. Sorted populations were ≥95% pure in all assays as determined by flow cytometry. For a negative control effectors, bead purified CD8+ T cells from naive Y10 mice were used. As target cells for these assays, CD8-depleted splenocytes that were either 1% dimethyl sulfoxide (DMSO) treated (control targets) or peptide pulsed with 25 µM D^bGagL peptide in 1% DMSO for 1 h at 37 °C were used. These target cell and effector cell populations were then placed in a 2-h in vitro cytotoxic killing assay at a 1:4 (10,000:40,000 cells) or 1:10 (10,000:100,000) target:effector ratio (T:E) for 2 h in the substrate following recommendations from the PanToxiLux Kit (OncoImmunin, Inc.). The samples were then immediately analyzed by flow cytometry for substrate fluorescence. For data quantification, the "background" level of substrate fluorescence from the DMSO control targets was subtracted from the CTL-killing level of the gag-peptide-loaded targets for each individual sample.

**In vivo killing assay.** The target cells for these assays were CD8-depleted splenocytes from C57/BL6 RRID: IMSR_JAX:000664 WT mice or CD47−/− RRID: IMSR_JAX:003173 mice on the C57/BL6 background were either 1% DMSO treated (control targets) or peptide pulsed with 25 µM D^bGagL peptide in 1% DMSO for 1 h at 37 °C. All target cells were then labeled with 375 ng/mL Deep Red (Invitrogen) and then differentially labeled with either 4 or 40 µM carboxyfluorescein succinimidyl ester (Invitrogen) or either 2.5 or 25 µM CellTrace^TM violet (Invitrogen). Naive control or chronically infected recipients were given an i.v. adoptive transfer of 5 × 10^6 cells of each subset at approximately 25:25:25:25 ratio, as confirmed by flow cytometry of the Time Zero sample. Spleens from

recipient mice were analyzed by flow cytometry after 6 h. The percentage of killing of each population of FV-pulsed cells was calculated as follows: 100 − ([% peptide pulsed in infected divided by % unpulsed in infected) divided by (% peptide pulsed in uninfected divided by % unpulsed in uninfected)] × 100).

**In vivo anti-PD-L1 blockade.** Y10 mice chronically infected with FV were injected i.p. every other day for 7 or 10 total injections with 250 µg functional grade 10F.9G2 (BioXCell). Control mice were concurrently given 250 µg rat IgG (BioXCell). Tissues were harvested the second day following the final injection.

**RNAseq.** RNA was isolated as described above for the in vitro suppression and reverse transcriptase–PCR assays. Total RNA was purified using phenol–chloroform extraction followed by RNeasy MinElute Cleanup Kit as per the manufacturer's instructions. cDNA libraries were prepared using the Ovation RNA-Seq system V2 by Nugen, Nextera DNA Library Preparation Kit for Illumina, and Nextera dual index (i7 and i5) adapter sequences. RNAseq was performed by the Stanford Functional Genomics Facility (Illumina NextSeq). Computing for this project was perfomed on the Stanford Sherlock cluster. Stanford Functional Genomics Facility extracted and generated FASTQ files for each sample, distinguished by the Nextera dual index adapters. Raw reads were trimmed for base call quality (phred ≥ 21) and adapter sequences using Skewer[78]. Processed reads were aligned to mm10 and read counts were generated using STAR 2.5.3a[79]. The R package "DESeq2" was used to normalize read counts, perform differential gene expression analysis, and generate the heat map. Transcripts were considered differentially expressed if they had a Benjamini–Hochberg adjusted p value <0.1.

**HCV cohort.** PBMC, plasma, and serum were studied in 15 HCV-infected patients. Ten patients underwent at least one previous treatment with interferon, the other five were treatment naive. PBMC, plasma, and serum were collected in ten non-infected patients as a control. Patients provided written informed consent for research testing that complied with the ethical regulations under protocol 13859 by the Stanford University Institutional Review Board. The characteristics of the patients are shown in Table 1.

**Phospho-CyTOF sample processing and staining.** Cryopreserved PBMCs stored at −180 °C were thawed in warm RPMI medium supplemented with 10% fetal bovine serum, benzonase, and a penicillin–streptomycin mixture (complete RPMI). Cells were transferred into serum-free RPMI medium containing 2 mM EDTA and benzonase, incubated with cisplatin for 1 min, and immediately quenched with four volumes of complete RPMI. Then one million cells per sample were transferred into complete RPMI and rested for 30 min at 37 °C. Following this rest period, cells were fixed in PBS with 2% PFA at room temperature for 10 min. Cells were then washed 2× with CyFACS buffer and barcoded using platinum- and palladium-labeled CD45 conjugates[80]. Following barcoding, samples were combined for surface marker staining, performed at room temperature for 1 h. Subsequently, cells were washed and permeabilized in MeOH at −80 °C overnight. The next day, cells were washed and incubated with the intracellular cytokine cocktail at room temperature for 1 h. DNA stain was performed for 20 min with iridium (191/193) in PBS with 2% PFA at room temperature. Finally, cells were washed 2× with CyFACS buffer and then twice with MilliQ water before data acquisition on the CyTOF2 instrument. Data was de-barcoded and manually analyzed on Cytobank (cytobank.org).

**In vitro stimulation of human CD8 T cells.** Ninety-six-well flat bottom tissue culture plates were coated with anti-human CD3 (1ug per ml) and anti-human CD28 (3ug per ml) in PBS for 2 h. One million PBMCs were then plated in each

well and stimulated (or left unstimulated in T cell media of RPMI containing supplementation with 50 units/mL IL-2 from Peprotech) for 5 days. Cells were then stained and analyzed by flow cytometry.

**Linear regression modeling**. For the estimation of regression coefficients, we iteratively conducted multiple linear regressions with the scalar-dependent variables set as the median expression of each marker in the major PBMC cell subsets and the explanatory variables set as age, sex, history of previous IFN treatment, history of cirrhosis, history of transplantation, sofosbuvir treatment regimen, HCV genotype, and HCV infection status. Regression coefficients with values different from zero at a false discovery rate threshold of $q < 0.05$ were considered significant.

**Reporting Summary**. Further information on research design and experimental design is available in the Nature Research Reporting Summary linked to this article.

## Data availability

The RNAseq data that support the findings of this study have been deposited in Sequence Read Archive with the project accession code SRP173611. The remaining data that support the findings of this study are available from the corresponding author upon reasonable request. A reporting summary for this Article is available as a Supplementary Information file.

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

## Acknowledgements

We thank Dr. Carrie Long, Dr. Clayton Winkler, and Dr. Catharine (Katy) Bosio for critical reading of the manuscript. We would also like to thank Anita Mora and Austin Athman for figure preparation and L.M.M would love to thank Flogging Molly for inspiration. This work was supported by: The Intramural Research Program of the National Institute of Allergy and Infectious Diseases, National Institute of Health, USA, Project ZA1000753; Virginia and D.K. Ludwig Fund for Cancer Research; Robert J. Kleberg, Jr and Helen C. Kleberg Foundation; AML grant R01CA086017; and the PCBC from NIHLB U01HL099999; and U19AI109662. M.C.T. and Y.Y.Y. were supported by the Stanford Immunology Training Grant 5T32AI007290 and M.C.T. was also supported by the NIH NRSA 1 F32 AI124558-01. L.B.T. was supported by the Stanford Diversifying Academia, Recruiting Excellence Doctoral Fellowship. E.A.P. was supported by F30DK099017 and the Stanford Medical Scientist Training Program. The funders had no role in study design, data collection and analysis, decision to publish, or preparation of the manuscript.

## Author contributions

L.M.M. and M.C.T. conceived and performed experiments, analyzed and interpreted data, and wrote the paper. L.B.T., A.B.C., R.J.M., M.M.S., C.L.A., R.S., M.M., Y.Y.Y., G.G., E.A.P., A.A., B.F. and A.M.N. performed experiments and analyzed data. J.S.G., M.M.D., and S.M.K. provided intellectual input and supervised experiments. I.L.W. and K.J.H. conceived experiments, interpreted results, and wrote the paper.

## Additional information

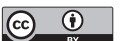

