## [Peer Review File · Nature Communications]

Reviewers' comments:

Reviewer #1 (Remarks to the Author):

1. Figure 1: it is not made clear how was SIRPa identified as coexpressed with PD1: "Dunnetts multiple comparisons test" implies many comparisons but we are not told how coexpression was tested nor what other genes were included in the search/what the threshold criteria for coexpression/significance were. The data on coexpression is not shown either. This risks undermining the central finding that SIRPa is coexpressed with PD1, from which the remainder of the manuscript flows.
2. The expression level of SIRPa is critical to the underlying argument (in particular because we need to understand why no-one else has found it on T cells) - Fig 1B should show raw data (FACS scatterplot) and not just summary MFI. The discussion should directly address why others have not identified expression of SIRPa.
3. Cross-reactivity within the SIRP family has previously been highlighted as a problem in identifying expression, for example: https://www.annualreviews.org/doi/full/10.1146/annurev-immunol-032713-120142?url_ver=Z39.88-2003&rfr_id=ori%3Arid%3Acrossref.org&rfr_dat=cr_pub=pubmed. How confident are the authors of their antibody's specificity? This makes point (2) all the more important and multiple antibodies/comparison with other family receptors is important to make this point.
4. There is not a clear coexpression pattern of SIRPa and PD1 as is claimed (Fig1C); PD1hi tend to be SIRPa+ but this is not the same as coexpression
- 5 Two panels in Fig C appear not to come from Panel B (14dpi/chronic infection: there appear to be more cells in B = fewer in C, the opposite of the panel below that is being expanded).
6. "only activated (CD11a+) CD8+ T cells expressed SIRPa during any phase of FV infection (Fig.1C) and non-activated cells remained negative (Fig.1D)" = this is directly contradicted by the data shown. Fig1D shows a small % of both Dex+ (mostly SIRPa-ve) and Dex-ve/Sirpa+ cells though no quantification is given. For this reason, also cannot safely conclude that "Sirpa induction required cognate TCR activation" as is concluded on p8. Also don't show expression on Dex-CD11a+ cells. Would expect both SIRPa+ and SIRPa- here if TCR stimulation is indeed important/essential (as there will be TCR stimulation of antigen specific cells responding to other FV peptides).
7. There is significant SIRPa mRNA expression in gdT but not in T cells.... .how do they explain lack of apparent mRNA expression in public databases, even of activated T cells? Expression on gdT shouldn't confound their current result but should be discussed.
8. It appears that there is no increase in SIRPa expression with division during chronic infection (Fig2H) - p value would confirm, not currently tested.
9. Conclusion at the end of p8 (donor cells from same naive pool show different SIRPa increase when transferred into either acute or chronic FV infection) requires a test of significance, ie comparing black dots in Fig2D v those in 2H. As this is the principal comparison of interest it would make sense for these to be plotted together.
10. The selection of individual divisions is unclear. x1 division is labeled at extreme right of dilution trace suggesting all cells have divided at least once. No pre-transfer control shown to indicate where '0' divisions actually is.

11. The proposed phenotypic description is inconsistent and misleading: increased memory (CD62L) and terminal differentiation markers (KLRG1) are inaccurately described within the text. "CD62L....intermediate onto majority of SIRPa+ CD8 T cells". This statement misses the point that there is a very significant difference in expression compared to the SIRPa- subset. "neither... showed high expression of the terminal differentiation marker KLRG1": this also misses the point that there is a very significant higher expression on SIRPa+ cells. Description of the data here is misleading and doesn't accurately reflect the data shown.

12. Fig3 contains histograms from a single expt without evidence of replication or tests of significance. Unclear how to interpret the data as a result.

13. Fig4. not sure why need to use adoptively transferred cells for this. Fig 1 suggests there is a large endogenous FV-specific Dex+ population in acute/chronic infection..... there are concerns over the use of large numbers of adoptively transferred cells given their subsequent homeostatic expansion that can skew their phenotype (see below).

14. Their GSEA analysis (with substantial overlap of both positive and negative immune regulators) highlights the inadequacies of database pathway annotation but adds little else (although none of the enrichment results are actually shown). There is excessive emphasis and overinterpretation of individual genes at the top of their differential lists, especially as they are not clearly T cell expressed genes. For example, the 2nd differential is an Fc receptor not known to be expressed on T cells either.... 1st is a myeloid receptor. The theme of identifying expression of genes not known to be T cell expressed is risky and consequently requires greater evidence to support its validity.

15. SIRPa+ cells are proposed to be PD1 coexpressing (Fig1), yet have evidence of increased cytotoxicity in both acute and chronic infections.... this implies that PD1hi cells have higher cytotoxicity in both contexts and are not actually functionally exhausted... this is confusing and contradictory with their central narrative. Are the Dex+ cells in chronic FV infection actually exhausted? The evidence presented in Fig 5 would suggest not (no difference in acute/chronic, irrespective of PD-1 or SIRP staining). The missing data on cytotoxicity from acutely infected mice might help address this, but it would still appear to be inconsistent with the GZMB/CD107a staining which implies similar degranulation.

The models used in Fig5 and earlier Figures are different (adoptive transfer vs endogenous response measurement), with the cytotoxicity model requiring adoptive transfer rather than using endogenous cells. This could explain differences, however the disparity is both problematic and confusing.

16. Line 215: figure labelling appears incorrect. Line 215 should read Fig6C, D presumably.

17. Figure 6 describes a complex experiment and consequently it is necessary to be absolutely clear in its representation. Specifically, a clear description of what is meant by 'virus specific killing' and hence what is being compared in 6E (and, therefore, represented and summarised in 6C,D) is missing. Some description is included deep in the methods but the formula for %killing should be made more accessible and the figure labeled to indicate its derivation as the figure is uninterpretable without it.

Also, there is clearly a difference in killing in the CD47KO chronically infected mice (6D) (peptide loaded v non). This suggests that at least part of killing is CD47 independent and this should be taken into account in the calculation of virus-specific killing. It doesn't appear that this is currently the case.

18. The adoptive transfer of large numbers of T cells has been shown to impact substantially on their surface phenotype (e.g. <https://www.nature.com/articles/nri2251>). It would be reassuring to know that the surface phenotype is not a by-product of the comparatively large numbers transferred.

19. The phenotype used to defined 'exhausted' cells taken during human HCV infection (CD57/CD28) uses different markers from that used previously in the manuscript (PD1+) and conflates 'terminal differentiation' with exhaustion. CD39 has been described as a useful surface marker of exhaustion in human cells and would be a clearer choice. Scatterplots to show the raw data would also be reassuring: these are Ag-specific CD8 populations and presumably very small populations. It would be reassuring to see the data as small differences in small populations can result in statistical significance without clear biological relevance.

20. The SIRPa + cells expand following PDL1 block but it isn't clear from the data whether they are the source population for this expansion. By comparison, a Tbet/Eomes subpopulation of exhausted cells has been clearly described to underpin the post-checkpoint expansion (see <https://www.nature.com/articles/nri3862> and references within). Comparison with this population is an obvious omission from the current manuscript.

Reviewer #2 (Remarks to the Author):

Hasenkrug and colleagues report the expression of signal response peptide 1a on anti-viral CD8 T cells during chronic viral infections. The upregulation of SIRPa appears to be a generalized phenomenon and occurs following both LCMV and FV infections of mice as well as during HCV infection. The patterns of SIRP1a expression by CD8 T cells resembles that of PD-1 which is known to be expressed by exhausted T cells and attenuates their function. During chronic infections the SIRP1a expressing and non-expressing cells are distinct, with SIRP1a expression associated with the more proliferatively active and functional subset of cells. These cells also appear to be the subset that is responsive to PD-1 checkpoint blockade. The studies are logical, informative, and well described. Strengths of the study include the identification of SIRP1a subsets in multiple systems (FV, LCMV, and HCV) and species (mice and humans). Also, the detection of SIRP1a on CD8 T cells is novel as it is typically expressed by non-lymphocytes. It is also notable that the authors show that SIRP1a influences the targeted killing activities of anti-viral CD8 T cells towards CD47 (the SIRP1a-ligand) expressing cells. Although there are strengths and novel aspects to the report, there are some issues, primarily with significance that limit enthusiasm, as outlined below.

1. A concern with this this report is significance. In essence the study describes another molecule expressed by exhausted T cells and adds to the catalog of markers are that are associated with these cells during persistent infections. The studies are legitimate, appear well performed, and the report is a pleasure to read, but given the current status of the field I am doubtful that this represents a major breakthrough in our understanding of exhaustion. Perhaps the report could be strengthened by determining why SIRP1a is expressed by a subset of exhausted cells and/or more fully exploring the consequences of its expression.

2. Several reports (Ahmed, Yu, and Ye, amongst others) have now shown that CXCR5 and TCF-1 expressing CD8 T cells play a critical role in containing persistent infections and represent the subset of exhausted cells that are responsive to checkpoint blockade therapies. It would be appropriate to ascertain the relatedness of the SIRP1a-expressing CD8 T cells to these exhausted follicular subsets.

3. Since the SIRP1a CD8 T cells appear more functional it would be of interest to determine if the presence, absence, or expansion of these cells change the viral loads in the infected mice, and are having a biological impact in vivo.

Reviewer #3 (Remarks to the Author):

This is an interesting study of SIRPa which emerged from a screen of genes which appear with a similar trajectory to PD-1 in LCMV infection. The authors provide evidence that SIRPa expression is linked to a sub-fraction of cells with distinct functions. Since SIRPa has not been shown to be expressed on T cells this is a novel finding. It is not yet clear whether SIRPa is responsible for the distinct functionality of the cells.

1. For the discovery aspect, some context would be helpful. How many genes show a similar trajectory? This could be addressed using a correlation analysis of the existing datasets.
2. Similarly, what are the expression levels for comparison (by FACS) on myeloid cell types.
3. There does not seem to be any difference in gene expression for SIRPa at day 8, but a FACS difference is seen. Could the authors clarify this by showing the full timecourse and the FACS data? This could be shown with PD1 for comparison with the FLV experiments.
4. The phenotypes of SIRPa⁺ v – cells are interesting in Fig 2. Sustained CX3CR1 expression looks to be at an intermediate level which has been shown recently to be associated with sustained proliferative capacity in LCMV and also persistent MCMV infections. Some of the other enriched genes and functions are also similar to published datasets on expanded MCMV subsets (eg Bolinger et al 2015, Quinn et al 2015; . It would be worth probably comparing these datasets by GSEA (ideally staining of such MCMV specific populations could be informative). Even if a negative result it would be worth checking in light of the CX3CR1/KLRG1 combination.
5. The relative distribution of SIRPa cells would be informative– this comment is prompted by the finding of CCR6 and CXCR6 on the gene list. This may reflect redistribution to certain tissues such as the liver, but also by findings of proliferation of cells in the lymph node following PD1 blockade. This may also be relevant for the conclusions and interpretation.
6. The enhanced killing of CD47 ko cells is interesting and suggests a functional interaction that is significant. However, it is not possible to tell if this is on the side of the target or the effector. In these experiments it should be possible to test if SIRPa⁺ cells show differential activation compared to SIRPa⁻ cells and also if they showed differential activation/cytokine release depending on CD47⁺ or – target.
7. The human data are not really in keeping with the rest of the paper. These are bulk cells, of which HCV-specific cells will be a tiny fraction. Also the patients are largely treated, so overall although there are some non-specific effects of HCV on other CD8 populations, I do not think it is possible to conclude so much from this part of the study. It would be really valuable to show that SIRPa is expressed on human cells under similar conditions. This could be done first off in vitro. Analysis of HIV⁺ patients' tetramer⁺ populations would be ideal if available. Also it would be important to see the FACS/Cytof staining data in such a study.
8. In the discussion could the authors describe the relevant phenotypes of SIRPa ko mice, which have been established for a while.

Reviewer #1 (Remarks to the Author):

1. Figure 1: it is not made clear how was SIRP α identified as coexpressed with PD1: "Dunnetts multiple comparisons test" implies many comparisons but we are not told how coexpression was tested nor what other genes were included in the search/what the threshold criteria for coexpression/significance were. The data on coexpression is not shown either. This risks undermining the central finding that SIRP α is coexpressed with PD1, from which the remainder of the manuscript flows.

There seems to be some confusion regarding the LCMV results in Figure 1A in which we identified *Sirp α* as having a similar expression pattern to *Pdcd1*, not as being coexpressed. Regarding the Dunnett's post-test: as stated in the legend, "SIRP α and PD-1 expression were analyzed by Dunnett's multiple comparisons test with each time-point compared to time zero." To address the issue of other genes and the threshold criteria, we have added a section on page 7 of the results and Supplemental Figure 1 showing that 1488 genes significantly correlated ($p < 0.05$) with *Pdcd1* expression in acute and chronic LCMV infection. When all of the genes are organized by order of correlation with the expression pattern of *Pdcd1*, *Sirp α* ranked in the 97th percentile. CD8+ T cell *Sirp α* expression significantly correlates with *Pdcd1* expression at a Pearson correlation coefficient of 0.516 and p-value of 0.001263.

Additionally, many of the genes that correlated with SIRP α expression from the publicly available microarray data also were significantly upregulated in

PD1⁺SIRPα⁺ CD8 T cells specific for FV gag. We appreciate the suggestion of adding in this analysis and have also added an additional Supplemental Table 3 with a Venn diagram now highlighting a transcriptional signature of genes that are upregulated along with SIRPα that are present in both LCMV and FV specific T cells.

2. The expression level of SIRPα is critical to the underlying argument (in particular because we need to understand why no-one else has found it on T cells) - Fig 1B should show raw data (FACS scatterplot) and not just summary MFI. The discussion should directly address why others have not identified expression of SIRPα.

We think that the Reviewer must be referring to Fig. 1A because the Friend Virus section already provided raw data. Thus, we have added representative raw flow cytometry data as a new Fig. 1B for LCMV. With regard to our novel identification of SIRPα expression on T cells, our guess is that previous testing for expression was predominantly done on naïve T cells rather than activated CTLs. Analysis of the publicly available data presented in Figure 1A clearly shows that others have indeed identified expression of SIRPα. In addition, SIRPα RNA is expressed during LCMV infection but not on naïve cells, just as we show for FV infections. We have added a statement to this effect to the first paragraph of the discussion as suggested. Furthermore, comparative analysis of SIRPα expression on T cells with that on macrophages has been added as supplemental Fig. 3.

3. Cross-reactivity within the SIRP family has previously been highlighted as a problem in identifying expression, for example: https://www.annualreviews.org/doi/full/10.1146/annurev-immunol-032713-120142?url_ver=Z39.88-2003&rfr_id=ori%3Arid%3Aacrossref.org&rfr_dat=cr_pub=pubmed.

How confident are the authors of their antibody's specificity? This makes point (2) all the more important and multiple antibodies/comparison with other family receptors is important to make this point.

Cross reactivity between SIRP gene products has been an issue with the human-specific antibodies and the question of specificity is certainly important. That said, the mAb p84 we use for flow cytometry is specific for mouse SIRPα and we

have added a short section to the materials and methods to address this issue: "mAb p84 specificity: Signal regulatory proteins (SIRP α and SIRP β in the mouse) are known to be expressed on neurons, hematopoietic stem cells and myeloid cells including macrophages, monocytes, granulocytes and dendritic cells¹. Dendritic cells, macrophages and monocytes from mice with targeted SIRP α gene disruptions completely lose reactivity with anti- SIRP α mAb p84 even though their SIRP β expression is not affected. These results indicate specificity for SIRP α without cross reactivity for SIRP β ²." Mice do not express a SIRP γ gene.

4. There is not a clear coexpression pattern of SIRP α and PD1 as is claimed (Fig1C); PD1hi tend to be SIRP α + but this is not the same as coexpression

Both panels in what is now Fig. 1D (due to the additional of the requested raw data for LCMV) definitively show that the vast majority of single cells expressing SIRP α also express PD-1, thereby meeting the definition of coexpression, which is the simultaneous expression of two or more genes by a cell. In addition, we have added Pearson Correlation plots in Figure 2A,E, to show the significant dual expression profiles of PD-1 and SIRP α .

5 Two panels in Fig C appear not to come from Panel B (14dpi/chronic infection: there appear to be more cells in B = fewer in C, the opposite of the panel below that is being expanded).

A representative FACS plot with gating strategy is simply shown, whether the FACS plot is from the same mouse or not seems irrelevant. The percentage in each quadrant depicts the mean, with standard deviation in parentheses, of n=8 mice combined from two independent experiments.

6. "only activated (CD11a+) CD8+ T cells expressed SIRP α during any phase of FV infection (Fig.1C) and non-activated cells remained negative (Fig.1D)" = this is directly contradicted by the data shown. Fig1D shows a small % of both Dex+ (mostly SIRP α -ve) and Dex-ve/SIRP α + cells though no quantification is given. For this reason, also cannot safely conclude that "SIRP α induction required cognate TCR activation" as is concluded on p8.

Also don't show expression on Dex-CD11a+ cells. Would expect both SIRP α + and SIRP α - here if TCR stimulation is indeed important/essential (as there will be TCR stimulation of antigen specific cells responding to other FV peptides).

It is true that there is a smattering of cells in the SIRP α positive quadrant, but there are also antigen experienced cells (CD11a+) even in naïve mice that could easily account for them due to gate placement. In addition, quadrants rarely have “zero” events due to background noise. That said, we have used more conservative language in the description. And as suggested, which is a valid point, we have also added Supplemental Figure 3 to show SIRP α expression on the CD11a+ Dex- cells, which is very similar to the expression on the Dex+ cells. Since we do not know the precise peptide specificity of the activated Dex- cells we concentrated our further analysis on the DbgagL-specific CD8+ T cells that stain with our dextramer reagent.

7. There is significant SIRP α mRNA expression in gdT but not in T cells.... .how do they explain lack of apparent mRNA expression in public databases, even of activated T cells?

Expression on gdT shouldn't confound their current result but should be discussed.

We are not aware of any literature stating that gamma delta T cells express SIRP α , but would appreciate the reference. Both the endogenous and transgenic T cells that react with tetramers or hexamers are alpha beta T cells, bind viral peptides and interact with MHC class I H-2Db molecules. This has now been noted in the materials and methods section. Regarding the lack of SIRP α RNA in public databases, as stated in the first paragraph in the results, the RNA expression data for Figure 1A, new Supplemental Fig. 1 and new Supplemental Table 1 were obtained from a public database, Ref. 26. (Crawford, A., Angelosanto, J. & Wherry, E. Whole mouse genome analysis of LCMV-specific CD4 and CD8 T cells throughout acute and chronic LCMV infection. *NCBI Gene Expression Omnibus* <https://www.ncbi.nlm.nih.gov/geo/query/acc.cgi?acc=GSE30431> (2012).). Thus, there is not a lack of mRNA expression in the public databases.

8. It appears that there is no increase in SIRP α expression with division during chronic infection (Fig2H) - p value would confirm, not currently tested.

There was a slight but significant increase in the proportion of cells expressing SIRP α between cell division 2 (mean = 23.5) and cell division 7 (mean = 32.83) P = 0.0241. Linear regression analysis has now been applied to analyze changes in SIRP α expression through cell division in both acute and chronic infections and the slopes of the lines have been compared and are statistically different (P =

0.0088). The linear regression and statistical comparisons have been added to the results and figure legend.

9. Conclusion at the end of p8 (donor cells from same naive pool show different SIRP α increase when transferred into either acute or chronic FV infection) requires a test of significance, ie comparing black dots in Fig2D v those in 2H. As this is the principal comparison of interest it would make sense for these to be plotted together.

Thank you for the suggestion. Linear regression analyses were used to draw the lines in Fig. 2D ($R^2 = 0.6148$, $P = 0.0369$) and Fig. 2H ($R^2 = 0.8837$, $P = 0.0053$). The difference between the slopes of the lines for acute ($m = 5.781$) vs chronic ($m = 1.836$) was very significant, $P = .0088$.

10. The selection of individual divisions is unclear. x1 division is labeled at extreme right of dilution trace suggesting all cells have divided at least once. No pre-transfer control shown to indicate where '0' divisions actually is.

Freshly labeled pre-transfer cells are too bright to be compared with post-transfer as all the cells show a major loss of MFI upon transfer into the animal. That said, the zero division intensity is not important for the main point of SIRP α and PD-1 expression with proliferation over time. We have added to the legend: "but the numbers are only relative as the zero division expression of CellTrace was not evident". In addition, we have also used more conservative language in the results section, for example, by removing "four to five cell divisions" in the section heading and stating "after cell division".

11. The proposed phenotypic description is inconsistent and misleading: increased memory (CD62L) and terminal differentiation markers (KLRG1) are inaccurately described within the text. "CD62L...intermediate onto majority of SIRP α + CD8 T cells". This statement misses the point that there is a very significant difference in expression compared to the SIRP α - subset.

"neither... showed high expression of the terminal differentiation marker KLRG1": this also misses the point that there is a very significant higher expression on SIRP α + cells.

Description of the data here is misleading and doesn't accurately reflect the data shown.

The descriptions of the phenotypes were certainly not meant to be misleading. Representative flow histograms along with average MFIs for each marker are shown in Figure 3 and it is possible for each reader to interpret the data. The comparison between SIRPα⁺ and SIRPα⁻ subsets for CD62L and KLRG1 expression has been added as suggested and we have re-worded the text more carefully.

12. Fig3 contains histograms from a single expt without evidence of replication or tests of significance. Unclear how to interpret the data as a result.

Histograms from one representative mouse are given to provide the reader with a picture of the data, but within the figure legend we stated: “The mean percent positive for each marker (the vertical dashed line delineates positivity) as well as the average geometric mean fluorescence intensity (MFI) from one experiment with 4 mice is given. Results are from one of three independent experiments with similar results (with n=8 additional mice). ns $p > 0.05$, * $p \leq 0.05$, ** $p \leq 0.01$, *** $p \leq 0.001$, **** $p \leq 0.0001$ (unpaired t tests).”

13. Fig4. not sure why need to use adoptively transferred cells for this. Fig 1 suggests there is a large endogenous FV-specific Dex+ population in acute/chronic infection..... there are concerns over the use of large numbers of adoptively transferred cells given their subsequent homeostatic expansion that can skew their phenotype (see below).

The use of the Thy1 genetic marker on the transgenic cells allowed us to sort the cells without stimulating them by crosslinking the TCRs with dextramers. The use of the transgenic cells also allowed us to obtain sufficient cells from single mice to avoid pooling cells from multiple mice, thus allowing us to compare the SIRPα⁺ and SIRPα⁻ cells from the very same mice. Fig. 2 demonstrates that, like endogenous cells (Fig. 1B), TCR Tg cells express SIRPα during FV infection and that the SIRPα⁺ subset percentages in acute and chronic infection were similar to the percentages in the endogenous populations.

14. Their GSEA analysis (with substantial overlap of both positive and negative immune regulators) highlights the inadequacies of database pathway annotation but adds little else (although none of the enrichment results are actually shown).

There is excessive emphasis and overinterpretation of individual genes at the top of their differential lists, especially as they are not clearly T cell expressed genes. For example, the 2nd differential is an Fc receptor not known to be expressed on T cells either.... 1st is a myeloid receptor. The theme of identifying expression of genes not known to be T cell expressed is risky and consequently requires greater evidence to support its validity.

In response to the criticism of excessive emphasis of individual genes at the top of the differential list, especially those known as myeloid-specific, we have eliminated that discussion. Although the pathway annotations in current databases are admittedly incomplete, the pathway analysis performed is current state of the art and is adequate to show that activation and cell division pathways are overexpressed in SIRP α ⁺ CD8 T cells compared to SIRP α ⁻ CD8 T cells. Furthermore, as we now point out in the text discussing Fig. 5, the direct *ex vivo* flow cytometric analysis of SIRP α ⁺ CD8 T cells support the findings of the GSEA and ToppFun analyses as do the phenotypic analyses in Fig. 3. The data are all consistent.

15. SIRP α ⁺ cells are proposed to be PD1 coexpressing (Fig1), yet have evidence of increased cytotoxicity in both acute and chronic infections.... this implies that PD1^{hi} cells have higher cytotoxicity in both contexts and are not actually functionally exhausted... this is confusing and contradictory with their central narrative. Are the Dex⁺ cells in chronic FV infection actually exhausted? The evidence presented in Fig 5 would suggest not (no difference in acute/chronic, irrespective of PD-1 or SIRP staining). The missing data on cytotoxicity from acutely infected mice might help address this, but it would still appear to be inconsistent with the GZMB/CD107a staining which implies similar degranulation. The models used in Fig5 and earlier Figures are different (adoptive transfer vs endogenous response measurement), with the cytotoxicity model requiring adoptive transfer rather than using endogenous cells. This could explain differences, however the disparity is both problematic and confusing.

The central narrative is that during so the called “exhaustion” of chronic infection, not all T cells are dysfunctional and there exists a subset that retains some function, a subset characterized by expression of SIRP α . In response to the Reviewer’s comments, we have now added new data comparing *in vitro* CTL killing by SIRP α ⁺ CD8 T cells and SIRP α ⁻ CD8 T cells from acutely infected mice in

Figure 5G. Although the SIRP α^+ CD8 T cells from acutely infected mice killed targets with more efficiency than the SIRP α^+ CD8 T cells from chronically infected mice ($p = .0079$), as in chronic infections, the SIRP α^+ subset had significantly better killing than the SIRP α^- subset. We have also added data on TCF-1 in Figure 5H&I expression showing that the SIRP α^+ CD8 T cells express lower levels of TCF-1 than SIRP α^- CD8 T cells, a phenotype associated with a less exhausted phenotype^{3,4}.

16. Line 215: figure labelling appears incorrect. Line 215 should read Fig6C, D presumably.

Thank you, the labeling has been corrected.

17. Figure 6 describes a complex experiment and consequently it is necessary to be absolutely clear in its representation. Specifically, a clear description of what is meant by 'virus specific killing' and hence what is being compared in 6E (and, therefore, represented and summarised in 6C,D) is missing. Some description is included deep in the methods but the formula for %killing should be made more accessible and the figure labeled to indicate its derivation as the figure is uninterpretable without it.

Also, there is clearly a difference in killing in the CD47KO chronically infected mice (6D) (peptide loaded v non). This suggests that at least part of killing is CD47 independent and this should be taken into account in the calculation of virus-specific killing. It doesn't appear that this is currently the case.

The definition of virus-specific killing has been added to the figure legend as suggested. Indeed, there is some CD47-independent, virus-specific killing in the CD47KO mice as the Reviewer notes. We state, "Thus, SIRP α -CD47 ligation was not required for cytolysis *in vivo*, but it significantly enhanced cytolysis..."

18. The adoptive transfer of large numbers of T cells has been shown to impact substantially on their surface phenotype (e.g. <https://www.nature.com/articles/nri2251>). It would be reassuring to know that the surface phenotype is not a by-product of the comparatively large numbers transferred.

The surface phenotypes (Fig. 3) were determined on endogenous cell subsets. The transcriptional profiles were done on transferred cells and were consistent with the endogenous cell upregulation of FASL, TIM3, CD44, LAG3, CD40, KLRG1, CD122 (IL2Rb) and Ki67 on the SIRP α + subset of CD8+ T cells. The complete list of all significantly upregulated genes is now shown in supplemental table 1.

19. The phenotype used to defined 'exhausted' cells taken during human HCV infection (CD57/CD28) uses different markers from that used previously in the manuscript (PD1+) and conflates 'terminal differentiation' with exhaustion. CD39 has been described as a useful surface marker of exhaustion in human cells and would be a clearer choice. Scatterplots to show the raw data would also be reassuring: these are Ag-specific CD8 populations and presumably very small populations. It would be reassuring to see the data as small differences in small populations can result in statistical significance without clear biological relevance.

Unfortunately, we did not have CD39 in our CYTOF panel, but we have now included FACS data of one of the HCV infected patients in Supplemental Figure 4A to demonstrate that we can confirm SIRP α expression on both CD57+ and CD57-CD3+CD8+ T cells from these patients. We have also adjusted our description of these T cell populations in the text.

20. The SIRP α + cells expand following PDL1 block but it isn't clear from the data whether they are the source population for this expansion. By comparison, a Tbet/Eomes subpopulation of exhausted cells has been clearly described to underpin the post-checkpoint expansion (see <https://www.nature.com/articles/nri3862> and references within). Comparison with this population is an obvious omission from the current manuscript.

The SIRP α positive and negative subsets did not have significantly different expression levels of Tbet, EOMES, CTLA4, or Bcl2 and this has been added to the discussion along with the comparison to the lineages described in the above references.

Reviewer #2 (Remarks to the Author):

Hasenkrug and colleagues report the expression of signal response peptide 1a on

anti-viral CD8 T cells during chronic viral infections. The upregulation of SIRP α appears to be a generalized phenomenon and occurs following both LCMV and FV infections of mice as well as during HCV infection. The patterns of SIRP1 α expression by CD8 T cells resembles that of PD-1 which is known to be expressed by exhausted T cells and attenuates their function. During chronic infections the SIRP1 α expressing and non-expressing cells are distinct, with SIRP1 α expression associated with the more proliferatively active and functional subset of cells. These cells also appear to be the subset that is responsive to PD-1 checkpoint blockade. The studies are logical, informative, and well described. Strengths of the study include the identification of SIRP1 α subsets in multiple systems (FV, LCMV, and HCV) and species (mice and humans). Also, the detection of SIRP1 α on CD8 T cells is novel as it is typically expressed by non-lymphocytes. It is also notable that the authors show that SIRP1 α influences the targeted killing activities of anti-viral CD8 T cells towards CD47 (the SIRP1 α -ligand) expressing cells. Although there are strengths and novel aspects to the report, there are some issues, primarily with significance that limit enthusiasm, as outlined below.

1. A concern with this report is significance. In essence the study describes another molecule expressed by exhausted T cells and adds to the catalog of markers that are associated with these cells during persistent infections. The studies are legitimate, appear well performed, and the report is a pleasure to read, but given the current status of the field I am doubtful that this represents a major breakthrough in our understanding of exhaustion. Perhaps the report could be strengthened by determining why SIRP1 α is expressed by a subset of exhausted cells and/or more fully exploring the consequences of its expression.

Significance:

1. SIRP α is not simply another marker added to the catalog of markers associated with exhausted cells. The significance of SIRP α is that it specifically marks the subset of these cells that retain cytolytic function. Although the reviewers tended to focus on the exhausted cells during chronic infection, we showed that SIRP α also marks the cytolytic cells during active acute infection. These results, along with the in vitro and in vivo CTL studies indicate an important functional role for SIRP α .
2. The discovery of SIRP α on activated, cytolytic CD8⁺ T cells is also significant because it is not supposed to be there. In the blood, SIRP α is only supposed to be

expressed on macrophages and rare hematopoietic stem cells. There are a multitude of papers stating that SIRP α is not expressed on T cells and the literature needs to be corrected. This erroneous information can lead to bad mistakes. For example, a gating strategy to remove macrophages from an analysis of T cells by gating out all SIRP α ⁺ cells would unwittingly remove activated T cells of interest. In addition, therapeutics targeting SIRP α on macrophages could unwittingly target CTL as well. The information presented here will open up new studies on the role of SIRP α in T cells compared to macrophages.

3. Biological processes are highly regulated and cells do not waste energy with the random expression of proteins. Thus, the appearance of SIRP α on the subset of cytolytically active CD8⁺ T cells during acute and chronic infections likely has important biological significance. One reason why SIRP α is expressed appears to be that interactions with CD47 molecules on target cells contribute to cytolytic killing, as we demonstrate in Fig. 6. However, there may be other important functions for this molecule as well. SIRP α expression is not only on exhausted cells but also on cytolytic cells that are highly functional during acute infections. We believe that further investigation of SIRP α functions will be of high interest once this discovery is reported.

4. Although the CD8⁺ T cells that expand following PD-1 blockade have been associated with certain surface markers, SIRP α is unique in that it marks the functional cells expanding during immunotherapeutics.

2. Several reports (Ahmed, Yu, and Ye, amongst others) have now shown that CXCR5 and TCF-1 expressing CD8 T cells play a critical role in containing persistent infections and represent the subset of exhausted cells that are responsive to checkpoint blockade therapies. It would be appropriate to ascertain the relatedness of the SIRP1 α -expressing CD8 T cells to these exhausted follicular subsets.

We have now added analysis of TCF-1 staining in Fig. 5H&I, which shows that the SIRP α ⁺ subset has significantly more expression of TCF-1 compared to the SIRP α ⁻ subset. This result is consistent with the SIRP α ⁺ subset having a less exhausted phenotype.

3. Since the SIRP1 α CD8 T cells appear more functional it would be of interest to determine if the presence, absence, or expansion of these cells change the viral

loads in the infected mice, and are having a biological impact in vivo.

The reviewer raises an important issue, but we don't yet have the tools to do depletion studies targeting SIRP α because none of the antibodies we have tested are depleting. We are working on that issue and also on developing conditional knockouts of SIRP α on CD8+ T cells. That said, the expression of CD107a by virus-specific SIRP α + CD8+ T cells, as well as *in vivo* cytotoxicity certainly suggests a degree of biological control of the virus *in vivo*.

Reviewer #3 (Remarks to the Author):

This is an interesting study of SIRP α which emerged from a screen of genes which appear with a similar trajectory to PD-1 in LCMV infection. The authors provide evidence that SIRP α expression is linked to a sub-fraction of cells with distinct functions. Since SIRP α has not been shown to be expressed on T cells this is a novel finding. It is not yet clear whether SIRP α is responsible for the distinct functionality of the cells.

1. For the discovery aspect, some context would be helpful. How many genes show a similar trajectory? This could be addressed using a correlation analysis of the existing datasets.

As detailed in the response to Reviewer 1, we now show that 1488 genes significantly correlated ($p < 0.05$) with *Pdcd1* expression in acute and chronic LCMV infection. Additionally, *Sirpa* expression significantly correlates with *Pdcd1* expression at a Pearson correlation coefficient of 0.516 and p-value of 0.001263. When all of the genes are organized by order of correlation with the expression pattern of *Pdcd1*, *Sirpa* ranked in the 97th percentile. A graphical representation of this correlation analysis for both *Sirpa* and *Pdcd1* is now provided as a Supplementary Figure 1.

Supplemental Fig. 1

2. Similarly, what are the expression levels for comparison (by FACS) on myeloid cell types.

We have now added the requested comparison as a flow cytometry histogram overlay as Supplemental Fig. 2.

3. There does not seem to be any difference in gene expression for *SIRPα* at day 8, but a FACS difference is seen. Could the authors clarify this by showing the full timecourse and the FACS data? This could be shown with *PD1* for comparison with the FLV experiments.

The *SIRPα* gene expression was drawn from the publicly available database while the FACS data was from our group. The experimental variation of when expression differences were noted between LCMV strains could be from a number of factors such as dosages, mouse age etc. We are not trying to make a major point here regarding differences between the LCMV strains. Our main

point is that SIRP α expression is observable both at the RNA and protein levels in activated but not naïve cells.

4. The phenotypes of SIRP α ⁺ v – cells are interesting in Fig 2. Sustained CX3CR1 expression looks to be at an intermediate level which has been shown recently to be associated with sustained proliferative capacity in LCMV and also persistent MCMV infections. Some of the other enriched genes and functions are also similar to published datasets on expanded MCMV subsets (eg Bolinger et al 2015, Quinn et al 2015; . It would be worth probably comparing these datasets by GSEA (ideally staining of such MCMV specific populations could be informative). Even if a negative result it would be worth checking in light of the CX3CR1/KLRG1 combination.

We have examined SIRP α expression in two different mouse infections and a human infection in three different laboratories and find upregulation on activated CD8⁺ T cells. We agree that it would be interesting to investigate more infections but we feel this is outside the scope of our study and best left to the experts in other viral infections to follow up on the findings presented in our manuscript.

5. The relative distribution of SIRP α cells would be informative– this comment is prompted by the finding of CCR6 and CXCR6 on the gene list. This may reflect redistribution to certain tissues such as the liver, but also by findings of proliferation of cells in the lymph node following PD1 blockade. This may also be relevant for the conclusions and interpretation.

This is an interesting point and we are actively investigating SIRP α expression on various cell types and tissues but we feel this is beyond the scope of this study.

6. The enhanced killing of CD47 ko cells is interesting and suggests a functional interaction that is significant. However, it is not possible to tell if this is on the side of the target or the effector. In these experiments it should be possible to test if SIRP α ⁺ cells show differential activation compared to SIRP α ⁻ cells and also if they showed differential activation/cytokine release depending on CD47⁺ or – target.

The enhanced killing was actually on the wt target cells compared to the CD47KO cells and indeed suggests a functional interaction. We intend to follow up on the

biochemical basis of this function but, again, we feel this is beyond the scope of the current study.

7. The human data are not really in keeping with the rest of the paper. These are bulk cells, of which HCV-specific cells will be a tiny fraction. Also the patients are largely treated, so overall although there are some non-specific effects of HCV on other CD8 populations, I do not think it is possible to conclude so much from this part of the study. It would be really valuable to show that SIRP α is expressed on human cells under similar conditions. This could be done first off in vitro. Analysis of HIV+ patients' tetramer+ populations would be ideal if available. Also it would be important to see the FACS/Cytof staining data in such a study.

In response to this comment we have added data in Supplemental Fig. 4B showing the upregulation of SIRP α on human proliferating CD8+ cells stimulated with anti-CD3 and anti-CD28 (shown below). We have also added FACS data from one of the patients profiled in the CYTOF data in Supplemental Fig. 4A as suggested, including comparison of CD8+ T cell expression of SIRP α with CD14+ macrophage expression. Much more extensive analyses are indeed warranted, but we think that it is important to show that SIRP α upregulation on CD8+ T cells occurs in humans as well as mice and worthy of further investigation by clinical scientists.

8. In the discussion could the authors describe the relevant phenotypes of SIRP α ko mice, which have been established for a while.

A discussion of the phenotypes of SIRP α mutant and KO mice has been added as suggested. “Mice with genetic inactivation or mutation of SIRP α have numerous abnormalities including impairment of: phagocyte migration⁵ dendritic cell homeostasis⁶, bone cell differentiation⁷, kidney function⁸, and IL-17 and IFN γ production by lamina propria cells⁹. Phagocytes from SIRP α mutant mice also have enhanced respiratory bursts¹⁰”

- 1 Dietrich, J., Cella, M., Seiffert, M., Buhring, H. J. & Colonna, M. Cutting edge: signal-regulatory protein beta 1 is a DAP12-associated activating receptor expressed in myeloid cells. *Journal of immunology* **164**, 9-12 (2000).
- 2 Washio, K. *et al.* Dendritic cell SIRP α regulates homeostasis of dendritic cells in lymphoid organs. *Genes Cells* **20**, 451-463, doi:10.1111/gtc.12238 (2015).
- 3 Wu, T. *et al.* The TCF1-Bcl6 axis counteracts type I interferon to repress exhaustion and maintain T cell stemness. *Sci Immunol* **1**, doi:10.1126/sciimmunol.aai8593 (2016).
- 4 Zhou, X. *et al.* Differentiation and persistence of memory CD8(+) T cells depend on T cell factor 1. *Immunity* **33**, 229-240, doi:10.1016/j.immuni.2010.08.002 (2010).
- 5 Alvarez-Zarate, J. *et al.* Regulation of Phagocyte Migration by Signal Regulatory Protein-Alpha Signaling. *PloS one* **10**, e0127178, doi:10.1371/journal.pone.0127178 (2015).
- 6 Saito, Y. *et al.* Regulation by SIRP α of dendritic cell homeostasis in lymphoid tissues. *Blood* **116**, 3517-3525, doi:10.1182/blood-2010-03-277244 (2010).
- 7 Koskinen, C. *et al.* Lack of CD47 impairs bone cell differentiation and results in an osteopenic phenotype in vivo due to impaired signal regulatory protein alpha (SIRP α) signaling. *The Journal of biological chemistry* **288**, 29333-29344, doi:10.1074/jbc.M113.494591 (2013).
- 8 Takahashi, S. *et al.* SIRP α signaling regulates podocyte structure and function. *Am J Physiol Renal Physiol* **305**, F861-870, doi:10.1152/ajprenal.00597.2012 (2013).
- 9 Kanazawa, Y. *et al.* Role of SIRP α in regulation of mucosal immunity in the intestine. *Genes Cells* **15**, 1189-1200, doi:10.1111/j.1365-2443.2010.01453.x (2010).
- 10 van Beek, E. M. *et al.* SIRP α controls the activity of the phagocyte NADPH oxidase by restricting the expression of gp91(phox). *Cell Rep* **2**, 748-755, doi:10.1016/j.celrep.2012.08.027 (2012).

Reviewers' comments:

Reviewer #1 (Remarks to the Author):

Thanks to the authors for addressing the points raised. I have no further comments on the points not listed below.

1. Thanks to the authors for the clarification provided, this is helpful. Need to confirm the significance threshold in additional SuppFigure1 is corrected for multiple comparisons. Still not being completely transparent though - not sure I understand the distinction between having 'a similar expression pattern' and 'coexpression'..... I think they mean similar longitudinal expression pattern over time.... in which case showing expression of each as a function of time would be helpful.

The correlation analysis in Supp Fig 1 also still doesn't help the reader understand how the authors came to investigate SIRPa instead of the other 585 genes showing stronger/more significant correlation with PDCD1. Clarification of this process would be helpful.

2. In Fig 1B the authors show summary SIRPa MFI values showing a modest shift in normally-distributed MFI values on LCMV-specific CD8 T cells at d8pi. I was interested to see an example of the scatterplot gating driving this modest shift in MFI on what may be a very small population (and hence very sensitive to small shifts) - ie as shown in Fig1D for FV. I cannot see such raw data in Fig1B (summary MFI) and SuppFig2 relates to FV infection. Supp Fig 3, showing relative expression of macrophages doesn't directly address this point (ie what are the relative protein levels on exhausted vs non-exhausted cells?). This is more than just pedantry on my part as the identification of SIRPa protein expression is central to the authors' argument. Others have identified SIRPa message, not protein. The key novel (and potentially controversial) finding here is of SIRPa protein expression, something others have not found significant levels of on T cells despite looking (SuppFig4 goes some way to helping this but comparable, clear demonstration on clone13/Armstrong-derived CD8 is an important omission. The authors 'guess' that previous testing was done on naive cells, but this can be checked. If this isn't the case, it must be explained or at least explicitly discussed.

6. It is of course reasonable to focus on the Dex+ Ag-specific cells for further analysis as the authors do. However, the data provided is far from sufficient to demonstrate the conclusion that SIRPa induction requires cognate pMHC:TCR interaction. I don't think softening this claim to a 'suggestion' is sufficient and it should be removed to avoid misleading the reader.

7. I agree that gDT expression is less relevant here. Relative gDT expression levels can be found through immgen (for interest): <http://www.immgen.org/databrowser/index.html>

14. My criticism here was not so much that the approach is inadequate but that the interpretation of the findings may be contradictory or confusing. The authors show significant enrichment of both positive and negative 'immune regulation' pathways from the GO classification. It can therefore only be concluded that transcripts altered are involved in 'immune regulation' without inferring direction of effect.

19. Supp Fig 4a is helpful. The expression level of SIRPa on unstimulated cells, however, appears higher than on most activated cells..... it therefore appears that SIRPa is downregulated on activation and retained on a small subset. This is not in keeping with the remainder of the data: is the baseline level of SIRPa in SuppFig4B incorrect?

Reviewer #2 (Remarks to the Author):

This revised report nicely catalogs the expression of signal response peptide 1a on both murine and human CD4 T cells following activation. The authors have responded to all of the points made during the prior round of reviews and have included new staining profiles for TCF-1. The novel aspect is that SIRP1a is expressed by activated CD8 T cells. It does remain uncertain regarding why this molecule is upregulated and how it is functioning to regulate the anti-viral response. Thus, some level of concern regarding significance remains. The other issue that was not well addressed is the biological impact of the SRP1a cells in vivo. It may be possible to investigate this point with adoptive transfer and challenge studies, however, this is not a major hold up and it is understood that certain reagents are not available.

Reviewer #3 (Remarks to the Author):

The authors have generally answered the questions OK but they somehow left more or less unaltered the HCV part.

According to the table, patients in all but one cases have been treated, with an SVR, so they do not have chronic HCV infection any more – unless the table is not clear and the samples were all taken before treatment (in which case the later SVR isn't really relevant).

The CytOF example is really helpful but current figure 8 is still quite hard to interpret - the subdivision into CD57+ doesn't really fit so well with the rest of the paper as these cells are not really exhausted eg with high PD1 expression, and CD57-CD28+ contains a mix of naïve and memory cells. The %s in the example - it seems to me - would be easier for the reader to understand the populations than the current y axis.

My suggestion would be that the authors clarify if the patients have HCV or not at the time of the blood draw. If they don't, as suggested by the table, then just provide the data from the CytOF/FACS study to show there are human T cells which express the molecule and some indication of which they are (it would be really helpful if the CytOF panel were described, but it should likely include regular memory markers). If they are chronically infected, it would still be easier to start off by showing that the molecule is expressed in healthy humans and then see if its expression is impacted somehow by chronic disease. Either way, if I understand the CytOF data presented, the molecule is detected in healthy controls as well as the HCV cohort so the text could reflect this.

Reviewer #1 (Remarks to the Author):

Thanks to the authors for addressing the points raised. I have no further comments on the points not listed below.

1. Thanks to the authors for the clarification provided, this is helpful. Need to confirm the significance threshold in additional SuppFigure1 is corrected for multiple comparisons.

We have confirmed significance for the correlation data for Supplemental Fig. 1 by multiple comparisons test using the Benjamini-Hochberg procedure ($p < 0.05$ and $FDR < 0.05$). The figure legend has been revised to state such.

Still not being completely transparent though - not sure I understand the distinction between having 'a similar expression pattern' and 'coexpression'..... I think they mean similar longitudinal expression pattern over time.... in which case showing expression of each as a function of time would be helpful.

The reviewer is correct, we do mean similar longitudinal expression pattern over time and we have now clarified the sentence. We show RNA expression of SIRP α and PD1 at 0, 1, 2 and 4 weeks post LCMV infection for Armstrong and Clone 13 strains in Fig. 1B.

The correlation analysis in Supp Fig 1 also still doesn't help the reader understand how the authors came to investigate SIRP α instead of the other 585 genes showing stronger/more significant correlation with PDCD1. Clarification of this process would be helpful.

We now further clarify our particular interest in SIRP α . SIRP α was of special interest because it had been shown to be important in innate immunity^{35 36} but had not been shown to be expressed on CD8⁺ T cells or any other adaptive immunity cells. Furthermore, the sustained expression of SIRP α on CD8⁺ T cells late after infection with Clone 13 suggested that it might identify an interesting subset of cells during exhaustion.

2. In Fig 1B the authors show summary SIRP α MFI values showing a modest shift in normally-distributed MFI values on LCMV-specific CD8 T cells at d8pi. I was interested to see an example of the scatterplot gating driving this modest shift in MFI on what may be a very small population (and hence very sensitive to small shifts) - ie as shown in Fig1D for FV. I cannot see such raw data in Fig1B (summary MFI) and SuppFig2 relates to FV infection. Supp Fig 3, showing relative expression of macrophages doesn't directly address this point (ie what are the relative protein levels on exhausted vs non-exhausted cells?). This is more than just pedantry on my part as the identification of SIRP α protein expression is central to the authors' argument. Others have identified SIRP α message, not protein. The key novel (and potentially controversial) finding here is of SIRP α protein expression, something others have not found significant levels of on T cells despite looking (SuppFig4 goes some way to helping this but comparable, clear demonstration on clone13/Armstrong-derived CD8 is an important omission. The authors 'guess' that previous

testing was done on naive cells, but this can be checked. If this isn't the case, it must be explained or at least explicitly discussed.

In response to the Reviewer's comments Fig. 1 B has been replaced with Flow cytometric contour plots of SIRP α expression on CD8⁺ T cells specific for Arm and Cl13 at 42 days post infection. "To verify protein expression, flow cytometry was used to analyze SIRP α expression on LCMV-specific transgenic CD8⁺ T cells at 42 days post-infection when CD8⁺ T cell responses to Armstrong would have contracted but responses to Clone 13 would be largely exhausted and express PD-1. A representative flow cytometry plot of the results shows that over 90% of the transgenic CD8⁺ T cells remaining after Armstrong infection were PD-1 low and did not express SIRP α (Fig. 1B). In contrast, over 95% of the transgenic CD8⁺ T cells remaining after Clone 13 infection were PD-1 high and a significant subset (mean = 14.9%) expressed SIRP α (Fig. 1B). The mean fluorescence intensity of SIRP α expression was significantly higher on the CD8⁺ T cells responding to the chronic Clone 13 strain compared to the acute Armstrong strain (Fig 1B, right).

Legend Figure 1. (B) Representative flow cytometry contour plots of Thy1.1-gated, adoptively transferred P14 CD8⁺ T cells at 42 days post-infection with the two LCMV strains is shown (left panels). Numbers in the upper right quadrant are means (N= 4 Arm, N=3 Cl13), P = 0.0029 by unpaired t test. The average MFI of SIRP α expression is shown in the right panel (P = 0.0088 by unpaired t test).

Our search of the literature shows no instances of anyone testing SIRP α protein expression on activated T cells. If the reviewer knows of such an instance and provides a reference we would be happy to examine and discuss the results. It is obviously our responsibility to ensure the validity of our results and we have answered all reviewer's questions to prove that we have accomplished that, both at the RNA level as shown in our paper and other cited papers (Refs 75, 76,77,78), and at the protein level, with the new data shown here and particularly in depth in the Friend virus model. We cite other instances where SIRP α RNA expression was seen in CD8⁺ T cells but we can't answer why nobody followed up on those findings.

6. *It is of course reasonable to focus on the Dex+ Ag-specific cells for further analysis as the authors do. However, the data provided is far from sufficient to demonstrate the conclusion that SIRPa induction requires cognate pMHC:TCR interaction. I don't think softening this claim to a 'suggestion' is sufficient and it should be removed to avoid misleading the reader.*

The statement has been removed as suggested.

7. *I agree that gdT expression is less relevant here. Relative gdT expression levels can be found through immgen (for interest): <http://www.immgen.org/databrowser/index.html>*

14. *My criticism here was not so much that the approach is inadequate but that the interpretation of the findings may be contradictory or confusing. The authors show significant enrichment of both positive and negative 'immune regulation' pathways from the GO classification. It can therefore only be concluded that transcripts altered are involved in 'immune regulation' without inferring direction of effect.*

The Reviewer is absolutely correct. We have modified the conclusions and discussion accordingly and discuss possibilities of positive signaling, negative signaling or no signaling through SIRP α , pages 18, 19, 20.

19. *Supp Fig 4a is helpful. The expression level of SIRPa on unstimulated cells, however, appears higher than on most activated cells..... it therefore appears that SIRPa is downregulated on activation and retained on a small subset. This is not in keeping with the remainder of the data: is the baseline level of SIRPa in SuppFig4B incorrect?*

Thank you very much for noticing this discrepancy. These data were sent to our graphics department as svg files as shown below to produce Fig. 4. Somehow during importation of the data into "Canvas" software to prepare Figure 4, the dot plot data became ungrouped from the axes and were shifted. Obviously, that is never supposed to happen and if it ever does it is our responsibility to catch it. It's disturbing both that it happened and that we didn't catch it. It was actually the data on the right that got distorted rather than the control. We can assure you that the data as now presented are correct and have not been altered or manipulated in any way. We have also gone back and checked all the figures to make sure all are correct. I personally sat and spoke at some length with the head of our graphics department, Anita Mora, (who made the mistake) to figure what happened and make sure that it never happens again.

Reviewer #2 (Remarks to the Author):

This revised report nicely catalogs the expression of signal response peptide 1a on both murine and human CD4 T cells following activation. The authors have responded to all of the points made during the prior round of reviews and have included new staining profiles for TCF-1. The novel aspect is that SIRP1a is expressed by activated CD8 T cells. It does remain uncertain regarding why this molecule is upregulated and how it is functioning to regulate the anti-viral response. Thus, some level of concern regarding significance remains. The other issue that was not well addressed is the biological impact of the SRP1a cells in vivo. It may be possible to investigate this point with adoptive transfer and challenge studies, however, this is not a major hold up and it is understood that certain reagents are not available.

The Reviewer is correct that we have not defined the function of SIRPa in CD8+ T cells. While we are working hard on the issue, it will likely take considerable time to finish that aspect of the research. We have added discussion regarding the function of SIRPa in activated CD8+ T cells to pages 18, 19, 20.

Reviewer #3 (Remarks to the Author):

The authors have generally answered the questions OK but they somehow left more or less unaltered the HCV part.

According to the table, patients in all but one cases have been treated, with an SVR, so they do not have chronic HCV infection any more – unless the table is not clear and the samples were all taken before treatment (in which case the later SVR isn't really relevant).

Thank you for pointing this out, we apologize for the confusion. The samples in the manuscript are all from before treatment. Treatment was irrelevant in this case and we have removed that column as well as the outcomes column from the manuscript methods description of the patient cohort.

The CytOF example is really helpful but current figure 8 is still quite hard to interpret - the subdivision into CD57+ doesn't really fit so well with the rest of the paper as these cells are not really exhausted eg with high PD1 expression, and CD57-CD28+ contains a mix of naïve and memory cells. The %s in the example - it seems to me - would be easier for the reader to understand the populations than the current y axis.

We have now added CYTOF data staining SIRP α on CD8 $^+$ T cells from HCV+ and HCV- patients, now Fig. 8A. and also flow cytometry data on supplemental Fig. 4A. "In CD8 $^+$ T cells from both HCV uninfected and infected patients, the main subset was SIRP α negative (Fig. 8A). However, in HCV patients there was a subpopulation of CD8 $^+$ T cells with increased expression of SIRP α (Fig. 8A and Supplemental Fig. 4A)".

Figure 8. Increased SIRP α expression on CD8 $^+$ T cells from hepatitis C virus (HCV) infected patients identifies a more activated phenotype. (A) Representative plots of CYTOF analysis of the median healthy donor and the median HCV patient donor in regards to SIRP α expression levels.

We have now tried to clarify the use of CD57 and CD28 as appropriate markers to look for SIRP α expression. "We analyzed CD57 and CD28 markers because chronic antigenic stimulation of human CD8 $^+$ T cells is associated with the down-regulation of

costimulatory CD28 and upregulation of CD57 and the CD57⁺ CD28⁻ subset is known to be increased in HCV patients {Manfras, 2004 #9423}. Furthermore, although the CD57⁺ CD28⁻ subset is heterogenous, it is generally associated with a reduced state of function and proliferation {Strioga, 2011 #9422}.” We hope that this satisfies the criticism. We had also stained these CYTOF samples for PD-1 but, unfortunately, we had a technical issue with the antibody at that time and we have no further clinical samples to analyze.

My suggestion would be that the authors clarify if the patients have HCV or not at the time of the blood draw. If they don't, as suggested by the table, then just provide the data from the CytOF/FACS study to show there are human T cells which express the molecule and some indication of which they are (it would be really helpful if the CytOF panel were described, but it should likely include regular memory markers). If they are chronically infected, it would still be easier to start off by showing that the molecule is expressed in healthy humans and then see if its expression is impacted somehow by chronic disease. Either way, if I understand the CytOF data presented, the molecule is detected in healthy controls as well as the HCV cohort so the text could reflect this.

Hopefully we have now clarified that the samples are comparing chronic HCV infection to healthy controls. We hope that the additional staining panels we have clarified that expression of SIRPa in CD3+CD8+CD57-CD28+ T cells is minimal in healthy controls but increased in HCV patients.

REVIEWERS' COMMENTS:

Reviewer #1 (Remarks to the Author):

Thank you to the authors for their time and efforts in answering my queries. My initial points have now been addressed and I have none further to add.

Reviewer #3 (Remarks to the Author):

The authors have addressed my comment.